# Visibility Enhancement and Fog Detection: Solutions Presented in Recent Scientific Papers with Potential for Application to Mobile Systems

**DOI:** 10.3390/s21103370

**Published:** 2021-05-12

**Authors:** Răzvan-Cătălin Miclea, Vlad-Ilie Ungureanu, Florin-Daniel Sandru, Ioan Silea

**Affiliations:** Automation and Applied Informatics Department, University Politehnica Timisoara, 300006 Timisoara, Romania; miclea_razvan@yahoo.com (R.-C.M.); florin.d.sandru@gmail.com (F.-D.S.)

**Keywords:** visibility enhancement, mobile systems, fog detection methods and systems

## Abstract

In mobile systems, fog, rain, snow, haze, and sun glare are natural phenomena that can be very dangerous for drivers. In addition to the visibility problem, the driver must face also the choice of speed while driving. The main effects of fog are a decrease in contrast and a fade of color. Rain and snow cause also high perturbation for the driver while glare caused by the sun or by other traffic participants can be very dangerous even for a short period. In the field of autonomous vehicles, visibility is of the utmost importance. To solve this problem, different researchers have approached and offered varied solutions and methods. It is useful to focus on what has been presented in the scientific literature over the past ten years relative to these concerns. This synthesis and technological evolution in the field of sensors, in the field of communications, in data processing, can be the basis of new possibilities for approaching the problems. This paper summarizes the methods and systems found and considered relevant, which estimate or even improve visibility in adverse weather conditions. Searching in the scientific literature, in the last few years, for the preoccupations of the researchers for avoiding the problems of the mobile systems caused by the environmental factors, we found that the fog phenomenon is the most dangerous. Our focus is on the fog phenomenon, and here, we present published research about methods based on image processing, optical power measurement, systems of sensors, etc.

## 1. Introduction

Adapting vehicle speed to environmental conditions is the main way to reduce the number of accidents on public roads [1]. Bad visibility caused by the weather conditions while driving proved to be one of the main factors of accidents [1]. The research from the last decade came with different features to help the drivers, such as redesigning the headlights by using LED or laser devices or improving the directivity of the beam in real time; with these new technologies, the emitted light is closer to the natural one [2]. In addition, they also introduced a new feature, auto-dimming technologies being already installed on most of the high-end vehicles [3]. In case of fog, unfortunately, this is not enough, and up until now, no reliable and robust system was developed to be installed on a commercial vehicle. There were approaches based on image processing by detecting lane marking, traffic signs, or hazards such as obstacles [4], image dehazing and deblurring [5], image segmentation, or machine learning methods [6,7]. Other methods are based on evaluating the optical power of a light source in direct transmission or backscattering, by analyzing the scattering and dispersion of the beam [8,9]. There are approaches that are using systems already installed on the vehicle such as ADAS (Advanced Driver Assistant Systems), LIDAR (LIght Detection And Ranging), radar, cameras, or different sensors [10,11,12] and even geostationary satellite approaches [13]. While imaging sensors output reliable results in good weather conditions, their efficiency is decreasing in bad weather conditions such as fog, rain, snow, or glare of the sun.

The biggest companies around the world are working these years to develop a technology that will completely change driving, the autonomous vehicle [14]. When this will be rolled out in public ways, the expectation will be for crashes to decrease considerably. However, let us think about how an autonomous vehicle will behave in bad weather conditions: loss of vehicle adherence, problems on vehicle stability, and maybe the most important fact is related to the decrease or lack of visibility: non-visible traffic signs and lane markings, non-identifiable pedestrian [15], objects or vehicles on its way [16], lack of visibility due to sun glare [17], etc. We have also the example of the autonomous vehicle developed by Google, which failed the tests in bad weather conditions in 2014. Now, the deadline for rolling out the autonomous vehicle is very close; 2020 was already announced by many companies, and they must find a proper solution for these problems because these vehicles will take decision exclusively based on the inputs obtained from the cameras and sensors or in case of doubts will hand over the vehicle control to the driver.

In the next decades, there will be a transition period; on the public roads, there will be autonomous vehicles but also vehicles controlled by the drivers; as drivers’ reactions are unpredictable, these systems will have to have an extremely short evaluation and reaction time to avoid possible accidents. Based upon this reasoning visibility estimation and the general improvement of visibility remain viable fields of study, we did a study on the state of the research for papers that use image processing as the means to estimate visibility in fog conditions, thus increasing general traffic safety.

Figure 1 presents an overview of the field, starting from the main methods from the state of the art, visibility enhancement (2), and fog detection (3), following by systems and sensors (4) that use the methods proposed in the first two subsections to detect visibility in adverse weather conditions and ending by presenting the human observer’s reactions in such conditions (5).

Basically, in the first category, the methods are based on image processing, while in the second one, they are based on optical power measurements or image processing. In the next sections, the most known and used methods from these two broad categories will be detailed. The goal of this work is to present the advantages but also the weaknesses of every method to identify new ways of improvement. Afterwards, as it is stated in the figure below, we propose a mix of methods with the scope of counterbalancing the shortages of a method with the other one. The final step will be to check if the results obtained from such a system are valid for human beings and additionally usable by autonomous vehicles.

The paper is structured as follows: we present visibility enhancement methods, fog detection methods, and sensors and systems that fulfill visibility measurements and detections in fog environment. Subsequently, we present some approaches that link the results obtained with an automatic system, presented above, and the human stimuli to understand their applicability now when humans need to take actions based on the system’s outputs but also for the future when an autonomous vehicle will make decision based on the same outputs. Afterwards, we present the conclusions of our work.

## 2. Visibility Enhancement Methods

In the last decade, there was a great interest in the area of improving visibility in bad weather conditions and especially in foggy conditions. The methods are based on image processing algorithms and can be split into two categories: image processing using a single input image (one of the first approaches was presented by Tarel and Hautiere in ([18]) and using multiple images ([19]) as input. Taking multiple input images of the same scene is usually impractical in several real applications; that is why single image haze removal has recently received much attention.

### 2.1. Basic Theoretical Aspects

This subsection will mathematically describe the main methods from the first category, visibility enhancement: Koschmieder Law and dark channel prior.

#### 2.1.1. Koschmieder Law

During his studies related to the attenuation of luminance through the atmosphere, Koschmieder noted that there is a relationship between the luminance of an object (L) found at a distance x from the observer and the luminance L_0_ close to the object:(1)$$L=L_{0}e^{-\sigma x}\;+L_{\infty }\left(1-e^{-\sigma x}\right)$$
where L_0_ is the luminance close to the object, $L_{\infty }$ is the atmospheric luminance, and σ is the extinction coefficient. By rewriting Equation (1), divining it by $L_{\infty }$ we will obtain:(2)$$C=(\frac{L_{0}-{\;L}_{\infty }}{L_{\infty }})\;e^{-\sigma x}=C_{0}e^{-\sigma x}$$
which is known as Koschmieder Law regarding the apparent contrast (C) of an object against the sky background at a certain distance x, considering the inherent contrast C_0_. This law is applicable only for daytime uniform luminance; for night conditions, we can apply Allard’s Law.

#### 2.1.2. Dark Channel Prior

This is a statistical method based on non-foggy images taken outdoors and uses a single foggy image as input. It is based on the statement that haze-free outdoor images have in almost all non-sky patches at least one channel with very low intensity for some pixels. In other words, the minimum intensity in such an area tends to zero. Formal, for an image J, can be defined:(3)$$J^{dark}(x)=min_{c\in \left\{R,\;G,B\right\}}(min_{y\in Ω\left(x\right)}\;\;(J^{c}\;(y)))$$
where $\;J^{c}$ is a color channel of image J, and $Ω\left(x\right)$ is a local area or window centered in x.

Except for the sky area from the image, J^dark^ intensity is very low; it is close to zero if J is an outdoor image without light. In this case, J^dark^ is called dark channel and the statistical analysis is known as “dark channel prior”.

### 2.2. Methods Based on Koschmieder Law

Koschmieder was one of the first researchers that treated the phenomenon of visibility degradation due to weather conditions. He studied the luminance attenuation in the atmosphere and proposed a law relating to the apparent contrast of an object against a sky background.

His was among the first approaches dedicated to transportation systems (started in 2005) from which were further derived numerous refinements belong to Hautière et al. [18,20,21,22,23,24], etc.

Based on the same principles, Negru et al. present in [25] and continuing in [26] a method of image dehazing, which calculates the attenuation of the luminance through the atmosphere. The goal of the work is to dehaze images taken from a moving vehicle and to inform the driver about the fog’s density and the adequate speed for those specific conditions. The first step of this approach is to apply a Canny–Deriche edge detector on the input image followed by estimations of the horizon line and inflection point of the image to indicate if fog is present. If haze is detected, then the extinction coefficient is calculated, visibility distance is estimated, and fog is classified considering its density (Figure 2). The method has good results for free, straight roads where the camera is not obstructed.

The authors deduced the visibility distance as:(4)$$d_{vis}=R_{t}v_{r}+\frac{{\left(v_{r}\right)}^{2}}{2gf}$$
where:*R_t_v_r_* of the equation represents the distance traveled during the safety time margin (including the reaction time of the driver), and the second term is the braking distance. This is a generic case formula and does not take into account the mass of the vehicle and the performance of the vehicle’s breaking and tire system.*R_t_* is a time interval that includes the reaction time of the driver and several seconds before a possible accident may occur.*g* is the gravitational acceleration, 9.8 m/s^2^*f* is the friction coefficient. For wet asphalt, we use a coefficient equal to 0.35.*v_r_* denotes the recommended driving speed.

Thus:(5)$$v_{r}=-gfR_{t}+\sqrt{g^{2}f^{2}R_{t}^{2}+2gfd_{vis}}$$

Later, Negru et al. proposed in [27] a contrast restoration method for an ADAS system. The method takes into consideration the exponential decay from a foggy image. The image is restored by estimating the atmospheric veil based on partitioning of the unity functions. The authors come with an ingenious method of filtering, by applying a median filter on the image’s columns, which increases the clarity of the reconstructed image (Figure 3). The advantages of this method are its performance in real-time applications and the possibility of using the algorithm on a mobile device, being a cost-efficient solution. The method does not offer reliable results for some specific cases, one example being the images with a constant depth of the observed scene.

A method based on a modified version of Koschmieder’s model is presented in [28], where the atmospheric effect of fog is modeled first. Afterwards, the atmospheric veil is estimated using dark channel prior, and then an exponential transformation is applied to it to improve the accuracy of the estimation. The authors assume that an input image is usually not affected in all the regions by fog, which is why it uses a nonlinear transformation not to affect the fog-free regions during the reconstruction process. Finally, every pixel from the image is reconstructed using the modified version of Koschmieder Law. Being a linear function of the number of input pixels, the complexity is low, which makes it proper to be used for real-time applications. Comparing the method with others from state of the art, from a quality point of view, it appears that the proposed method has better results in recovering details and edges, the colors of the output images being less affected in comparison to [18,29]. The assessment from quantity was performed considering only two parameters: ratio of visible edges added after the reconstruction process and the ratio of pixels that become saturated after the restoration process. The method has comparable results with the two methods mentioned above, the performances being related to the input hazy image.

An ADAS-based dehazing model is presented in [30], which continues the work started in 2012 by Tarel et al. in [23], where they exposed the idea to use a Head-up display (HuD) inside the vehicle, helping the driver to see behind the fog, by dehazing images in real time using Koschmieder model. During the experiments, the reaction time of the driver was tested to understand the usability of such an application. The results obtained after three experiments realized in the laboratory are only a starting point, the system needs to be tested on the roads in real conditions to prove its reliability and robustness. Such a system can be used in low fog conditions, but for dense levels of fog, the system is unable to help the driver.

Methods based on Koschmieder Law are applicable only in daytime scenarios, making them unusable for real-life automotive applications where the system offers reliable results 24 h/day.

### 2.3. Methods Based on Dark Channel Prior

One of the most influential papers in this field of removing haze from a single image was published by He et al. [31], which was based on the same principle described by Koschmieder. The presented method, dark channel prior, is based on the statistics of the haze-free outdoor images. The idea of this method is that at least one of the color channels has a very low intensity at some pixels (tends to zero) in most of the non-sky patches (Figure 4). Based on this assumption, it can estimate the haze thickness, and a high-quality haze-free image can be restored by the atmospheric scattering model. The advantages of the method are its simplicity and efficiency, but on the other hand, being a statistical method may not work for some specific images; in the case when the scene objects are similar to the atmospheric light, these objects will be ignored.

Yeh et al. in [32,33], based on the same idea mentioned above, presented two methods: pixel-based dark channel prior and pixel-based bright channel prior. They identified that the three major drawbacks in He’s paper—low accuracy, the computational complexity for big patches, and transmission map refinement—improved with the actual method. The proposed algorithm follows the next step: the atmospheric light is estimated using haze density analysis; then, using a bilateral filter, the transmission map is estimated and refined. The experiments show that the method presented in this paper has better results in recovering hazy images, achieving better color information and lower run time compared to He’s method ([31]). The performance related to color information makes the output images more natural compared to other methods from the state of the art. The dehazing results of this method were compared with the ones of the other three methods ([31,34,35]) from a visual quality point of view, being assessed by 10 subjects. The final score, after analyzing the dehazing results of 50 images, shows that the proposed method outperforms the others. The authors plan to extend the dehazing method to videos, with the idea of integrating this algorithm in a driving assistance safety device.

Huang et al. analyzed in [36] He’s work ([31]) and classified the drawbacks of this method in three categories—halo effects and color distortions in the recovered image plus insufficient transmission map—and developed a novel method desiring to fix these problems. The proposed method is based on three modules: depth estimation, color analysis, and visibility restoration. The experimental results of the actual approach are compared with He’s approach ([31]), considering four areas: refined transmission results, enhanced transmission result, results of haze removal, and quantitative evaluation. The analysis over these four areas proved that the proposed method outperforms other methods, being an efficient visibility restoration method proper to be applied in different weather conditions in a real environment. A comparison to [31] is done in Table 1.

The authors of [37] present a novel method where simultaneously a single hazy image is dehazed and the sharpness is enhanced. The transmission map algorithm is based on a fusion of transmission maps: a patch dark channel prior transmission map on one side and a single-point pixel one on the other side. Afterwards, a Gaussian kernel function is applied to improve the fusion. The main advantage of this method is its processing speed, due to the low complexity (linear function of the number of input image pixels) being adequate for real-time processing applications such as ADAS systems. It is also able to inhibit halos completely, improving visual quality. Comparing to [16], the method can provide more detailed information but has lower performance from dehazing point of view.

Zhang et al. propose in [38] a visibility enhancement method from a single hazy image. Based on the dimension reduction technique, the method uses a filtering approach (first proposed by Tarel in [39]), composed from a median filter and the truncated singular value decomposition to estimate atmospheric veil with dark channel prior to restore the haze-free image. Then, the method was compared with other methods from the state of the art from visual effects, speed, and objective evaluation criteria points of view. It turned out that the method is very fast due to the low complexity, but the results for heavy fog and far objects are not very reliable.

The foggy image enhancement method proposed in [40] follows the next steps: first, a dark channel computation is applied on the foggy image; then, the transmission ratio is estimated and adjusted. The atmosphere light is also estimated, while the last step of the algorithm is the gamma adjustment to get the final deblurred image. The method is compared with [31] from a dehazing performance and speed point of view, being inferior in the first category, the output images of the proposed method are unnatural but outperform in the second category, being twice as fast compared the other method.

One of the basic characteristics of a hazy image is the low contrast; that is why the main idea of the method proposed in [41] is to restore a hazy image by increasing the contrast. In addition to the contrast element, Kim et al. took also into consideration the loss of information in developing this optimized contrast enhancement algorithm valid for hazy images and video restoration. For static image dehazing, the algorithm consists of extracting the atmospheric light of an input image and then estimating the block-based transmission followed by a refinement of the transmission (Figure 4). A big advantage of the proposed method is that besides the applicability on static images, the algorithm was extended also for real-time videos by adding the temporal coherence cost and reducing computation complexity to avoid flickering artifacts, but this is still not possible to be implemented in application with limited computing resources, such as the actual system from the vehicles.

From the site http://mcl.korea.ac.kr/projects/dehazing/ (accessed on 24 April 2021), we can find many results of the project (and those in Figure 4), compared with processing including using other established algorithms. There is also video footage that was processed with the algorithm proposed by the authors. However, the effort is calculated, and for real-time application in the automotive sector, it not yet feasible.

During the experimental results, the method was compared with a few methods from the state of the art: Ref. [34] generates many saturated pixels by simply increasing the contrast of the restored image, Ref. [35] does not remove haze from some demanding regions, for instance near the horizon or around some objects, Ref. [32] changes color tones and produces halo artifacts, while the authors in [31] only considered the darkest pixel value for dehazing; it removes shadows from pictures. Going further in comparison with [31], the experiments show that the atmospheric light is estimated better with the proposed method, assures higher quality transmission maps, and truncates all the pixels (not only the dark ones); the overall results prove the higher dehazing performance of the proposed method.

Having the goal to improve image quality and increase the visibility of photos affected by haze, Ref. [46] proposes a method of recovering a single hazy image. The paper focuses on the fact that the dark channel prior algorithm tends to underestimate the transmission of bright areas. Therefore, starting from the atmospheric scattering method and then applying a weighted residual map and offset correction, an appreciable outcome is achieved. The experimental results demonstrate that image contrast was enhanced after applying the proposed algorithms, which led to more information being recovered and image clarity being improved. The combination of offset correction and residual map reduces color oversaturation and enhances details in dehazed images. Similar to other dehazing algorithms using the atmospheric scattering model, the presented method also suffers from the drawback of color shifting in defogged images.

Real-time processing is a must when discussing traffic safety features or autonomous driving. Focusing on that, Ref. [47] tackled the fact that dark channel prior has a high complexity, which makes the algorithm very demanding for real-time processing. The paper presents a Graphics Processing Unit (GPU) accelerated parallel computing method capable of real-time performances when removing haze from a high-definition video. To improve the performances, a filter method called a transposed filter combined with the fast-local minimum filter algorithm and integral image algorithm is used. Experimental results show that the proposed algorithms can process a 1080p video sequence with 167 frames per second, proving that the solution qualifies for real-time high-definition video haze removal. The video flicker problem is solved by the inter-frame constraint used to dynamically adjust the atmospheric light. The method drawback is that it requires extra hardware such as a GPU device, leading to an increase concerning the cost of production.

When discussing traffic safety features, real-time processing is very important; it being mandatory in such circumstances is the main subject of the paper [48]. Ngo et al. provide data about a single image haze removal algorithm and a specific hardware implementation that is facilitating real-time processing performances. The proposed approach mainly exploits computationally efficient processing techniques such as multiple-exposure image fusion, adaptive tone remapping, and detail enhancement. Compared to other state-of-the-art techniques, good performance is obtained by having low computational complexity. Compact hardware is used to handle high-quality videos at a rate higher than 25 frames per second. All datasets alongside the source code are available online for public use and could be used as a starting point for an autonomous vehicle.

Single image dehazing has made huge progress in recent years, but the focus was mainly on daytime. During the nighttime, the process is more challenging; due to multiple scattering phenomena, most daytime dehazing methods become invalid. The paper [49] proposes a novel unified nighttime hazy image enhancement framework that approaches both haze removal and illumination enhancement problems simultaneously. A big plus is the fact that most current daytime dehazing methods can be incorporated into nighttime dehazing tasks based on the proposed framework. The hazy image is decomposed into a halo layer and a scene layer to remove the influence of multiple scattering. Then, the spatially varying ambient illumination is estimated based on Retinex theory. The result is obtained by employing the classic daytime dehazing methods to recover the scene radiance and generate the dehazing result by combining the adjusted ambient illumination and the scene radiance. The framework is tested using various daytime dehazing methods (classic methods such as He’s dark channel prior), and a comparison from the performance point of view is performed proving the value of the work.

After analyzing the approaches from this chapter, we can conclude that the dark channel approach has some drawbacks, which make it unlikely to be used in the automotive field: dense fog images cannot be dehazed, and for some specific images, the method cannot apply, such as for example when objects are very close to the atmospheric light.

### 2.4. Image Segmentation Using Single Input Image

Zhu et al. in [50] present a mean shift-based algorithm, which improves a few issues of the traditional dehazing methods: where the sky is part of the image such as oversaturated images in [34], the time-consuming and limitation of using the method for gray-scale images in [35], and problems in handling sky images and computational complexity for [31,44]. The mean shift-based algorithm is composed of three steps: sky segmentation, re-refining, and restoring. The algorithm starts by computing the dark channel on the input image; then, on the same input image, a white balance correction is applied to reduce the influence of the color cast (after dehazing, there is a strong enhancement in saturation). On the processed image, the dark channel is re-calculated to get the coarse transmission map, and next, it is refined with guided image filtering (proposed in [44]). Afterwards, the sky regions are picked out using a shift segmentation algorithm to solve the problem of underestimation of the transmission. In the last phase guided image, filtering is applied again to smooth the transmission map, which is then dehazed obtaining the output dehazed image. From a qualitative point of view, the experiment results show that the proposed method improves the weaknesses of the approaches from the state of the art presented above. From a quantitative point of view, the method presented in this paper has almost the same results from noise production and image quality point of view but has lower results for dehazing effects in terms of contrast enhancement comparing with other approaches.

Methods based on the same idea of splitting the sky area from the rest of the image are presented in [51,52,53]. The assumption of the authors in [51] was that the sky patches are not important for the driver, so after applying dark channel on the hazy image, these sky patches can be masked (using Hough transform and edge detection methods); the output of this operation is an enhanced image and a decrease of the artifacts along the edges between the sky and road. In [52], the sky region is first segmented (quad-tree splitting and mean-shift algorithm) from the input hazy image, which is followed by a region-wide medium transmission estimation and refinement of the transmission map using a guided filter. Zhu et al. in [53] propose an algorithm where the sky region is first detected through brightness and gradient and then is split by the non-sky area to avoid the noise and color distortions caused by the sky area.

The algorithms for separating road and sky by region growing suffer from high computational costs and are therefore not optimal for a real-time application on an embedded system.

### 2.5. Image Segmentation Using Multiple Input Images

Some methods related to image segmentation (single image dehazing) were already presented, Refs. [50,51,52,53], where the sky was split from the rest of the image to avoid noise and distortions in the boundary area. In other approaches, this time having multiple input images, as presented in [54], the moving vehicles are segmented from the outdoor environment, even in bad weather conditions such as fog. Such a method requires a large number of frames to distinguish between the foreground and background, so the idea from this paper was to use the motion energy of the moving vehicle to differentiate them from the changing background. By applying a dynamic adaptive threshold, the false motion determined by the dynamic background is suppressed, and the computation efficiency is improved, making it feasible to be implemented in real-time applications. In comparison to other methods from the state of the art, the proposed method proved the robustness in reducing false motion and detecting moving vehicles in poor visibility conditions. One of the limitations of the proposed method is the case when two vehicles are very close; the algorithm detects them as being a single object, which is a fact that can be improved by using post-processing techniques. The method is likely to be used on highways as a static system that notifies the drivers about the visibility range.

Yuan et al. in [55] consider the foggy images being a convolution between the original image and the degraded function. The proposed method starts with a segmentation of the input image in blocks, which are then decomposed in background and foreground images, the latter one with sparse errors due to the movement. The next step is to build the local transfer function from deconvolution using the blocks with the lowest sparsity, with the main goal to create the global transfer function used to reconstruct a non-foggy image by deconvolution of the original image and the global transfer function. The results show that the proposed method provides clearer scenes with better visibility and more entropy information compared to other methods. The experiments using this method were performed on traffic images with simple scenes and with a low number of vehicles, it needs to be tested in a more complex situation to verify if the outputs are still reliable.

The method presented in [56] avoids the split between the background and the foreground of the image due to numerous disadvantages such as a continuous update of the image background due to changes in luminance, shadow effects, vibration in illuminance, and vehicle overlapping if these are very close to each other. The proposed method is based on an adaptive vehicle detection approach where vehicles are directly detected without involving the background operations. The first step is to normalize the input image by histogram extension (HE) to remove the impact of weather and light effects. Afterwards, the moving objects are dynamically segmented using the gray-level differential value method, and vehicles are extracted by merging broken objects or by splitting the incorrectly merged ones. Vehicle tracking, correlating with the existing information, and updating traffic parameters are the next steps, followed by an error compensation, which is useful for cases when some targets are missing. Comparing to previous methods, the processing time decreases, algorithms being applied only in the region of interest, working well in real time. The experimental results show that the method offers good results also for traffic jam conditions. The method was tested in different weather conditions (sunny, rain, sunrise, sunset, cloudy, snowy day), but it still needs to be evaluated in foggy weather conditions, which is the main area of interest for our research, making the method attractive for commercial applications.

### 2.6. Learning-Based Methods

Some very promising methods for visibility enhancement tasks, using a single image as input, are based on Neural Networks. This is a class from Machine Learning methods that allow solving complicated nonlinear functions. In the image processing area, Deep Neural Networks (DNN) are mostly used, which are Artificial Neural Networks (ANN) with many hidden layers of neurons between the input and output layers.

One of the pioneers in this field is Cerisan in [57], which won the German traffic sign recognition benchmark, being more precise in identifying objects than the human observers. In the visibility enhancement area, Hussain presents in [58] a method that has as an input a foggy image, models the fog function with deep neural networks, and outputs a fog-free image. The multiple hidden layers are useful in this case, in realizing a more efficient representation of the fog function. The network learns the function through some examples, basically by using input–output pairs of images. Initially, some random weights are used for the DNN, and the input pattern is propagated into the network, which produces a different output compared to the target. The error function of the network is a sum of the errors generated by every hidden layer; the solution proposed in this paper to minimize the error is a backpropagation algorithm that sends back into the network the error signal from the output layer; in this way, every node can calculate its own error introduced during the processing phase, and the nodes can update their input weights values. This process continues until the error becomes sufficiently small; basically, the output image generated by the network being acceptable is regarded as the target image. At this point, the input pattern is presented into the network. The algorithm is tested only on artificial images, which makes the job simpler as far as for such cases the patterns from such an image are more regular.

In [59], Singh proposed a method based on a particular case of DNN called Conventional Neural Networks (CNN); the method accepts both forward and backward propagation for error minimization. The CNN is implemented based on two primary layers: feature extraction and feature mapping. In [60], Cho et al. use a CNN as a classifier for visibility estimation; the network is trained using CCTV camera images captured in various weather conditions. To have a bigger number of images to train the network, the researchers proposed an algorithm that applies several augmentation techniques (rotation, flip, translate, zoom, region-zoom, etc.), so multiple unique images can be obtained from a single input image. Feature extraction uses two convolution layers; the first one produces 32 feature maps, and the second one produces 64 feature maps. The classification accuracy of the proposed method is about 80%; there is still room for improvement. One additional drawback is that only daytime images were used in the learning process.

To solve the fact that haze significantly reduces the accuracy of image interpretation, Ref. [61] proposes a novel unsupervised method to improve image clarity during the daytime. The method is based on cycle generative adversarial networks called the edge-sharpening cycle-consistent adversarial network (ES-CCGAN). Unlike most of the existing methods, this approach does not require prior information, since the training data are unsupervised. The focus of the study was on improving images captured by satellites, but the principles can be adapted to a vehicle environment. The presented experimental results prove the fact that the hazy image was recovered successfully and that the color consistency was excellent. The drawback of such an approach is that the performance depends on the training data, since many remote-sensing images are needed to train the algorithm.

Ha et al. presents in [62] a new dehazing technique. A residual-based dehazing network model is proposed to overcome the performance limitation in an atmospheric scattering model-based method. The proposed model adopted the gate fusion network that generates the dehazed results using a residual operator. The divergence between a dehazed and a clean image is reduced by analyzing the statistical differences via adversarial learning. Experiments were performed, and the results show that the method improves the quality of an image compared to other state-of-the-art approaches on several metrics. The proposed method was tested only in daytime scenarios, and it showed limitations when dense haze is present. In future works, the authors want to tackle the dense haze removal topic and to improve the current work.

It seems that learning-based methods are becoming more and more popular since many studies focused on this approach. The paper [63] proposes an unsupervised attention-based cycle generative adversarial network to resolve the problem of single-image dehazing. The novelty of the paper consists of an attention mechanism that can be used to dehaze different areas based on the previous generative adversarial network dehazing method. Different degrees of haze concentrations are targeted by this method while the haze-free areas are not changed. Training-enhanced dark channels were used as attention maps; in this manner, the advantages of prior algorithms and deep learning were combined. The presented experiments show the value of the work and demonstrate that the proposed technique can effectively process high-level hazy images and improve the clarity of the outcome. There are still some drawbacks to this approach, such as the fact that the dark channel is still erroneous for the area marked with haze. Ideally, the attenuation map of a haze-free image should be completely black; this is not yet achieved in the current work.

Focusing on learning-based solutions, Ref. [64] proposes a technique to recover clear images from degraded ones. A supervised machine learning-based solution is proposed to estimate the pixel-wise extinction coefficients of the transmission medium, and a compensation scheme is used to rectify the post-dehazing false enlargement of white objects. The focus is on a camera-based system being able to process images in real time. The fact that 4K videos can be processed at 30.7 frames per second in real time is a big gain of the presented work. Experimental results were performed to prove the superiority of this method over existing benchmark approaches. The source code and datasets are publicly available for further research.

During the nighttime, relying on light sources is not feasible because of inconsistent brightness and the cost of resources, especially when talking about a vehicle. The paper [65] proposes an autoencoder method to solve the overestimation or underestimation problem of transmission captured by the traditional prior-based methods. An edge-preserving maximum reflectance prior method is used to remove the color effect of hazy images. Then, it will serve as input for a self-encoded network to obtain the transmission map. Moreover, the ambient illumination is estimated through a guiding image filtering. The experimental results show that the proposed method can effectively suppress the halo effect and reduce the effectiveness of glow. The proposed method works well in keeping the edges of the image and suppress the halo effect. The drawbacks are that the color of the image changes after dehazing and that the estimation accuracy of ambient lighting and transmission map has a great influence on the quality of haze-free images.

Advanced learning-based methods, using deep learning, prove very good performances but for this, it needs a high amount of training images, in varying conditions, to learn the network; the performances of the network are limited by the amount of trained data used. Despite their many advantages such as the possibility of implementing different methods without many constraints, as it was observed in this chapter, versatility, and even price, cameras have also many drawbacks: they can be totally or partially blinded by other traffic participants or by weather conditions, providing erroneous results exactly at the moment when they are most needed.

## 3. Fog Detection and Visibility Estimation Methods

In the previous section, we mentioned Hautière and He as pilots for the field of image dehazing; now, one of the most relevant works for vision in the atmosphere is the work of Nayar and Narasimhan [42], which is based on reputed research of Middleton [66] and McCartney [67].

Most of the approaches for detecting fog and determining its density for visibility estimation are based on optical power measurements (OPM), but there are also image processing approaches. The basic principle of the methods from the first category is the fact that infrared or light pulses emitted in the atmosphere are scattered and absorbed by the fog particles and molecules, resulting in an attenuation of the optical power. Methods of detecting the attenuation degree are by measuring the optical power after the light beam passed a layer of fog (direct transmission) or by measuring the reflected light when the light beam is backscattered by the fog layer. Figure 5 provides an overview of optical power measurement methods.

### 3.1. Basic Theoretical Aspects

This subsection will mathematically describe two main methods from the fog detection through light scattering category: Rayleight scattering and Mie scattering.

#### 3.1.1. Rayleigh Scattering

Rayleigh scattering is applicable for light scattering where the size of the particles (x) from the atmosphere is much smaller than the wavelength (λ) of the light (x < λ/10). The intensity of the scattered radiation, I, can be defined as the product between the initial light intensity I_0_ and the Rayleigh scattering term S(λ, θ, h):
(6)$$I=I_{0}\;S(\lambda ,\;\theta ,\;h)=I_{0}\cdot \frac{\pi^{2}{\left(n^{2}-1\right)}^{2}}{2}\;\frac{\rho \left(h\right)}{N}\;\frac{1}{\lambda^{4}}\;\;(1+\cos^{2}\theta )$$
where λ is the wavelength of the input light, θ is the scattering angle, h is the position of the point, n is the refraction index, N is the density of the molecular number of the atmosphere, and ρ is the density level, which is equal to 1 at sea level and decreases exponentially with h.

The Rayleigh scattering equation shows how much light is scattered in a specific direction, but it does not indicate how much energy is scattered overall. For this, it must be taken into consideration the energy scattered in all directions:(7)$$\beta (\lambda ,\;h)=\frac{8\pi^{3}{\left(n^{2}-1\right)}^{2}}{3}\;\frac{\rho \left(h\right)}{N}\;\frac{1}{\lambda^{4}}$$
where β(λ, h) represents the fraction of energy lost after a collision with a single particle. This is known as the Rayleigh scattering coefficient or the extinction coefficient.

The initial Equation (4) that described Rayleigh scattering can be rewritten as follows:(8)$$S(\lambda ,\;\theta ,\;h)=\beta (\lambda ,\;h)\;\gamma (\theta )\;\to \;\gamma (\theta )=\frac{S\left(\lambda ,\;\theta ,\;h\right)\;}{\beta \left(\lambda ,\;h\right)\;}=\frac{3}{16\pi }\;(1+\cos^{2}\theta )$$
where the first term β(λ, h) controls the intensity of the scattering while the second one γ(θ) controls the direction of scattering. The last term is not dependent on the input light wavelength anymore.

Rayleigh scattering is applicable for particles whose size is less than 10% from the incident radiation wavelength. If the particle size becomes bigger than this value, the Mie scattering model can be applied to identify the intensity of the scattered light, which is the sum of an infinite series of terms, not just a simple mathematical expression such as in the Rayleigh model.

#### 3.1.2. Mie Scattering

According to ISO 13321:2009, the model is applicable for particles under 50 μm. Dust, pollen, smoke, and microscopic water drops that form mist or clouds are common causes of Mie scattering.

Comparing to Rayleigh scattering, for Mie scattering, the influence of input light wavelength is very low; the variation of the intensity is much stronger in the front direction compared to the rear one, and the difference increases with the particle dimension.

Mie theory is a theory of absorption and scattering of flat electromagnetic waves of uniform isotropic particles, having simple shapes (sphere, infinite cylinder) that are part of an infinite, dielectric, uniform, and isotropic environment.

The main scope of the theory is the calculation of efficiency coefficients for absorption (Q_a_), scattering (Q_s_), and extinction (Q_e_), which are connected through the following formula ([68]):Q_e =_ Q_a_ + Q_s_(9)

The scattering and extinction coefficients can be represented as infinite series, but for satisfactory convergence, the series shall not be longer than jmax = x + 4x^1/3^ + 2 (where x = 2πr/λ is the diffraction parameter):(10)$$Q_{s}=\frac{2}{x^{2}}\;\sum_{j=1}^{\infty }\left(2j+1\right)({\left|a{}_{j}\right|}^{2}+{\left|b{}_{j}\right|}^{2})$$
(11)Qe=2x2 ∑j=1∞2j+1Reaj+bj.

Re is the real part of the complex numbers a_j_ and b_j_:(12)$$a_{j}=\frac{\psi {}_{j}\left(x\right)\left[\frac{{\psi_{j}}^{\prime }\left(m_{\lambda }\;x\right)}{\psi_{j}\left(m_{\lambda }\;x\right)}\right]-m_{\lambda }\;{\psi_{j}}^{\prime }\left(x\right)}{\xi {}_{j}\left(x\right)\left[\frac{{\psi_{j}}^{\prime }\left(m_{\lambda }\;x\right)}{\psi_{j}\left(m_{\lambda },x\right)}\right]-m_{\lambda }\;{\xi_{j}}^{\prime }\left(x\right)};\;b_{j}=\frac{m_{\lambda }\;\psi_{j}\left(x\right)\left[\frac{{\psi_{j}}^{\prime }\left(m_{\lambda }\;x\right)}{\psi_{j}\left(m_{\lambda }\;x\right)}\right]-{\psi_{j}}^{\prime }\left(x\right)}{m_{\lambda }\;\xi_{j}\left(x\right)\left[\frac{{\psi_{j}}^{\prime }\left(m_{\lambda }\;x\right)}{\psi_{j}\left(m_{\lambda }\;x\right)}\right]-{\xi_{j}}^{\prime }\left(x\right)}.$$

The two coefficients a_j_ and b_j_ are called the Mie coefficients or the expansion parameters; these are expressed in terms of Riccati–Bessel functions Ψ_j_ (t) and ξ_j_ (t), which in their turn are expressed as Bessel functions of unintegrated order:(13)$$\Psi_{j}\;(t)=\sqrt{\frac{\pi t}{2}}\;J_{j+1/2}(t);\;\xi_{j}(t)=\sqrt{\frac{\pi t}{2}}\;J_{j+1/2}(t)+(-1{)}^{\mathrm{ni}}J_{-n-1/2}(t),\;i=\sqrt{-1}.$$

The absorption coefficient Q_a_ is determined based on the other two coefficients Q_e_ and Q_s_, using Formula (7).

### 3.2. Optical Power: Direct Transmission Measurement

The authors in [8] present laboratory measurements where fog is generated in a chamber and using a laser and an optical receiver, the fog influence (attenuation and absorption) on the optical laser beam is analyzed. The distance between the transmitter and the receiver is one meter, the fog environment is assured using a fog generator, the quantity of fog being controlled by the level of liquid used to generate it. In addition to these measurements, in the same chamber but without fog, the input power of the light source is varied to find a concordance between input power and different fog levels. The method is a good starting point to split fog into different categories by analyzing the input power or the optical output power, and the added value is the way the results are validated using an optical chart.

A method of estimating fog density based on the free space optical link attenuation, in real-life conditions this time, is presented in [69] by comparing the optical attenuation caused by bad weather conditions with the results obtained using standard meteorological equipment and with the ones obtained with a camera. The measurements are done in an area prone to fog, the observatory being at 836 m above sea level. The distance between the transmitter and the receiver is 60 m, having a simplex optical communication link at 1550 nm. The results of the optical system are closed to the ones obtained with the professional sensors. The work continues in [70,71], which present a link between attenuation caused by fog and visibility (definition from CIE):β = 3.91/V [km^(−1)],(14)
where β is the attenuation, V is visibility, the relation being valid for the wavelength λ = 550 nm. Comparison between the above-mentioned optical measurements and other methods are presented in the next subsection.

### 3.3. Optical Power: Backscattering Measurement

The authors in [9,72] present a fog sensor that measures the fog density based on the liquid water content (g/m^3^) from the atmosphere, which is measured considering the backscattering principle. The system is composed of an outdoor unit that sends and receives short infrared pulses. The amplitude and shape of the reflected pulses are analyzed, this being influenced by the liquid water content from the environment. The presence of fog is determined by comparing the computed fog density with a predefine threshold.

Using the principle of sending pulses in the medium and then measuring the time it takes to return, the light detection and ranging (LIDAR) systems [10,11] are able to detect particles down to a few μg/m^3^ with a spatial resolution of a few meters. The distance is calculated by multiplying the speed of light (0.3 m/ns) with the time of flight and then dividing by two due to the two ways covered by the beam. The LIDAR sends around 150,000 pulses per second, building up a complex map of the surface in front of it. Being a laser scanner device, it can be used also in other applications, such as weather conditions detection, which is an opportunity that is presented in [12,73]. It is possible to differentiate weather conditions from other objects considering the shape of the response that is compared with a predefined set of values. Fog particles, rain drops, and air pollutants have a flat and extended receiving echo, making them distinguishable from the ones coming back from a vehicle. In addition, from the shape of the response can be extracted information related to the absorption degree, having a reliable parameter for visibility estimation. The parameters that are considered during the evaluation are the reflected signal power and the backscattering coefficient. This system has a very big advantage because LIDARs are already installed on commercial vehicles, the data gathered from the LIDAR need only to be processed and used for this new feature. The validity of the results obtained with such a system still needs to be validated and confirmed in different weather conditions.

A method to detect precipitation and fog was proposed in [74], using a LIDAR ceilometer that analyzes backscatter data to estimate cloud height and attempting to detect the weather phenomena through machine learning techniques. Therefore, the backscatter data obtained from the LIDAR ceilometer is used as an input for the used algorithms. The precipitation detection shows potential, but unfortunately, the fog detection did not have promising results. Nevertheless, the paper can be used as a starting point for future research.

### 3.4. Image Processing: Global Feature Image-Based Analysis

Some other approaches for fog detection are using image processing methods, such as analyzing the global features of the input images. For example, [75] analyzes the power spectrum to detect fog conditions, which is the squared magnitude of the Fourier transform of the image, containing information about the frequencies in the image without considering any spatial information. To extract the features, first, a Han window is applied to avoid broadband components along the axis, followed by FFT. Before the classification, to get an accurate one, a two-step feature reduction is performed. Building these features, images can be classified afterwards as non-foggy and foggy images; for the latter category, the split can be done for different levels of fog (low fog, fog, dense fog). For fog scenes, the frequency components are concentrated near-zero frequency, while for non-fog scenes, there are much more high-frequency components. The results obtained after the experiments (image classification into different categories of fog) show an overall accuracy of 94% from the total number of tested images (44,000 images) in daytime conditions. During the experiments, all the road profiles were not covered, only three categories were considered: no fog, low fog, and dense fog. Another weakness of the method is that it can work only if the horizontal line is visible in the image; if other objects or vehicles surround the visibility, the results are erroneous. The last remark is that the method can be applied only in daytime conditions, which is a drawback that was improved in [76] where the method was extended for night conditions, the detection rate was improved (95.35% in daytime conditions and more than 99% for night conditions), and the method is more robust to variations. The method provides wrong results in detecting clear weather when high contrast appears in the image, which is understood as fog (e.g., oncoming vehicle, overtaking trucks, passing bridges, etc.).

The authors of [77] present an ADAS fog detection approach with high robustness to illumination changes by taking care of the frequency distribution (noise results in high frequencies) of the image blocks affected by fog (instead of analyzing the spatial domain); the computational costs are low comparing with methods from Section 2.2, and the goal of the authors is to achieve real-time requirements, which are essential for such a system. This method assumes that fog, being composed of fine water particles, diffuses light beams, which from an image processing point of view means that the edges of a foggy image are not as sharp as the ones of a non-hazy image, and visibility is reduced. The method starts with the calculation of the vanishing point from the input image; then, the power spectrum slope is calculated for the blocks around the vanishing point. By classifying (using naive Bayes classifier) the power spectrum slope parameters, a decrease in the visibility range is obtained (Figure 6). After analyzing more than 1100 images (around 15% natural images and the rest synthetic scenes), the detection rate for images affected by fog is more than 95% for small PSS (Power Spectrum Slope) blocks and decreases if the PSS block size is increased.

The tests were done in different light conditions, which is a very important aspect when we refer to a visibility detection system. On the other side, the algorithm was not tested in complex traffic situations, such as a crowded road with vehicles that can block the camera view, curves, or bridges. The algorithm, in this state, cannot be integrated into an ADAS system as far as it does not offer any other information besides that there is a foggy environment. Such a system shall inform the driver about the visibility distance, maybe about the speed that needs to be adapted considering the weather conditions, etc.

Asery et al. in [78] classify the hazy images based on their optical characteristics; the Gray Level Co-occurrence Matrix (GLCM) features are extracted after the RGB input image is split into three gray images. The three parameters considered to be the most important from GLCM are contrast, correlation, and homogeneity; these are used as classification parameters for support vector machine classifier. These three parameters are suitable for both synthetic images (accuracy of 97%) as well as natural images (accuracy of 85%). The results are better for synthetic images because the background remains the same for foggy and non-foggy images, which is a fact that is not valid in real life. The proposed method is compared with [79] for natural images, outperforming from an accuracy point of view.

In [80], Alami et al. considered only two parameters for fog detection: saturation and correlation. The selection of these parameters is based on the fog characteristics, color attenuation, and an increase in the white color. Fog detection is realized in two steps: first, the focus is on the vanishing point, which is detected with an edge-based algorithm; then, the straight lines, characterizing road objects such as borders, lane marking, etc., are detected using Hough Transform, the candidates’ vanishing points being the intersection points of the straight lines. In the second step, fog is detected by selecting a region centered at the vanishing point where the correlation and the saturation between the RGB channels is calculated for every pixel. For a foggy image, the region around the vanishing point is characterized by null saturation and high correlation. The method was tested only by synthetic images, so it still needs to be confirmed on natural images.

The backscattering method, for fog detection and visibility estimation, was used also in the image processing field, not only for optical power measurements. Gallen et al. patented a method ([81]) for visibility estimation based on fog detection in night conditions, which was further published in [82]. Fog is detected by analyzing the backscattering veil or halo that forms around a light source (e.g., headlamps, public lighting). The analysis was done empirically, by comparing a foggy image with a reference one with known visibility distance (and fog density). The authors agreed that this is not the most reliable solution, which is why for the future they plan to analyze the halo characteristics for detecting the fog.

The authors in [79] propose a histogram evaluation method; Zhang et al. present a comparison between image processing methods for visibility estimation by analyzing the input images’ features. There are seven histogram-based methods analyzed: Color and Edge Directivity Descriptor (CEDD), Edge Histogram Descriptor (EHD), Fuzzy Color and Texture Histogram (FCTH), Fuzzy Opponent Histogram (FOH), Joint Histogram Descriptor (JHD), Scalable Color Descriptor (SCD), and Simple Color Histogram (SCH). Support Vector Machines (SVMs) were used for classification, by employing Radial Basis Function (RBF) kernel and the Grid Search method for parameter optimization. The experiments used 321 images from three weather observation stations, which were captured with six cameras. During the evaluation, images were split into three categories—no fog, light fog, and heavy fog—and the methods’ classification is done by evaluating the accuracy of each of them by placing the images in the right category. The results show that JHD has the best performances followed by FCTH.

Neuronal Network methods are also used for fog detection not only in image enhancement. Some very novel methods in classifying images by analyzing their global features are based on Deep Neural Networks (DNN). Based on the backpropagation algorithm for minimizing the error, Pagani et al. present in [83] a two 5-layers neural network trained using an image captured by traffic camera spread around the Netherlands, with the main goal of identifying different categories of fog in the image. The authors propose a pre-processing phase before applying the DNN to unify the images from different cameras by reducing their dimensions to 28 × 28 pixels and by blurring them to avoid the presence of some specific pixels (e.g., specific information provided by a camera such as data, hour, location, etc.) that can be learned by the network. The set of features extracted are used as predictors to identify the fog density. After training the network using the H_2_O library, the method was used on some images obtained from traffic cameras, but the results were not satisfactory; the error rate was quite high. However, the development of the system is ongoing; the authors are planning to improve it by using additional training datasets with higher variety (day/night, different weather conditions, etc.).

Li et al. proposed in [84] a two CNN-based approach where the first CNN is used for visibility feature extraction, and based on these features, a generalized regression neural network (GRNN) is applied for intelligent visibility evaluation (approximating the function of visibility). The GRNN is a four-layer network, containing an input, a radial, a linear, and an output layer. For weather evaluation, only parts of the input image are used, the context and the image size used for the analysis influencing the accuracy of visibility prediction. The outputs of the method are not reliable at this point; the predicted accuracy is around 60%, the model being also influenced by the low amount of training datasets used to learn the network.

In [85], Chabaani et al. present a visibility range estimation method under a foggy environment. The method is proper to be part of a static system installed on highways or express roads, and it uses a single foggy image as input. The system has to differentiate the images in classes, starting from no fog to dense fog images. The authors proposed a three-layer neural network as a classifier, which was trained using a backpropagation algorithm. The network is learned using labeled examples and focuses more on global images’ features. The input layer represents the image features descriptor while the output layer represents the visibility range classes. Compared to the previous method ([83]), the actual method does not need a pre-processing phase, camera calibration, or information related to the distances in the depth map, being somehow generic for different types of images.

Five well-known dehazing algorithms (dark channel prior (DCP) [31], Tarel method [18], Meng method [86], DehazeNet method [87], and Berman method [88]) were compared [89]. Two images under different levels of fog were used for testing alongside their corresponding fog-free original image (captured in the visible and near-infrared). More details regarding the test images are provided in the “Spectral Image Database” section of the paper. From the quality measurement side, different commonly used metrics were selected from different categories (full-reference metrics, reduced-reference metrics, no-reference metrics). Among the used metrics are e Descriptor (the metric refers to the amount of new visible edges that were produced after dehazing [90,91], a higher e Descriptor value means better quality), Gray Mean Gradient (GMG, the metric refers to texture characteristics of the image [91,92], a higher GMG value means more visible edges), Standard Deviation (Std, the metric refers to the contrast of the image, a higher value means better quality), Entropy (a higher value mans a greater amount of information is contained by the image), Peak Signal to Noise Ratio (PSNR, compares two images, a higher value means that the images are more similar) and Structural Similarity Index Measure (SSIM, this metric refers to image quality from a human perception [93], the closer the value is to 1, the more similar the images are). Table 2 provides an overview of the comparison of these methods.

As presented in Table 2, the dark channel prior method excels when discussing edges visibility and contrast but lacks in similarities to the haze-free image, while DehazeNet and Tarel methods are on the opposite side. Berman and Meng methods provide a good average and all the metrics, performing well on most of the topics.

Apart from an objective classification from the metrics point of view, two surveys on 126 participants were conducted. In the first one, the subjects were asked to compare the haze-free images and the dehazed images. In the second one, the participants were asked to judge the dehazed images and the increase of the visibility of objects without taking into consideration the haze-free image. The results are presented in Table 3.

The human subjects consider that the Meng method produces the best results when discussing similarities to the original images, and the Berman method produces the best results when judging the increase in visibility of the objects presented in the image.

The results do not reveal a method that is the best in every aspect, but when using a dehazing algorithm, it is important to establish what metrics are of interest and based on that, a technique can be selected.

Visibility estimation methods are considered by our research team suitable for practical use to increase the safety of transport of any kind (road, maritime, airports), together with adequate maintenance of the road infrastructure (traffic management systems, markings, signaling, road, vehicles).

### 3.5. Visible Light Communications

Visible light communications (VLC) is a wireless technology suitable for data transmission using LEDs. This technology shows great potential for indoor applications, and now, it is proposed to be used in an intelligent transport system for vehicle-to-vehicle communications. Table 4 provides an overview of VLC patterns and potential applications. From the VLC perspective, diverse elements from traffic can be analyzed: infrastructure (traffic lights, traffic signs), vehicles in a junction (Head to Head communication, Head to Tail communication, Tail to Head communication, Left side communication, Right side communication), or parked vehicles (Tail to Tail communication, intelligent parking slots, parking spots that are on the side of the road). All of these situations can create events when a vehicle is starting to move, pedestrians also shall be taken into consideration. Some of these situations are taken into consideration in [94].

As stated, applications for V-VLC include intersection assistance, emergency breaking, intersection coordination, or parking assistance, all these applications having real benefits in the context of an intelligent transport system.

The paper [95] presents an experimental approach to analyze the effects of the fog phenomenon on optical camera-based visible light communications (VLC) in the context of an intelligent transport system. The authors used a real LED-based taillight and camera inside a fog chamber to simulate outdoor foggy weather conditions. A wide range of meteorological visibilities (5–120 m) is considered, several scenarios being tested (e.g., different values for modulation index of the signal). The obtained results show that the link is reliable up to 20 m meteorological visibility for a modulation index (MI) of 0.5. The results were better for a MI of 1, where the data transmission was reliable up to 10 m meteorological visibility.

Tian et al. present the effects of weather on Maritime VLC and how sea fog can directly affect signal transmission. As stated, the paper [96] focuses on sea fog and maritime communication, but the presented knowledge can be used in Vehicular VLC, too. To address the issue, the authors, provide a scattering spectrum analysis of fog sea particles and the attenuation of Maritime VLC. Taking into consideration the spectral data extracted from the LED, the spectral change of wide-spectrum LED at sea at a given distance is highlighted.

Wireless connectivity in vehicle-to-vehicle (V2V) and vehicle-to-infrastructure (V2I) is considered a worthy candidate in the development of an intelligent transport system (ITS). Therefore, its capabilities in an outdoor environment, affected by rainy and foggy weather conditions, are analyzed by the paper [97]. The V2V link is considered as a function of distance under specified weather conditions. This expression is used to determine the maximum communication distance that can guarantee a certain error rate. The results are presented in Table 5.

Table 5 shows that the maximum achievable distance is obtained for clear weather by 2-PAM (Pulse Amplitude Modulation) (72.21 m). As expected, the value is reduced by weather conditions to 69.13 m for rainy weather, 52.85 m for foggy weather with visibility V = 50 m, and 26.93 m for foggy weather with visibility V = 10 m. Another observable fact is that with the modulation size increase, the maximum distance for reliable transmission is reduced. The presented results only capture the direct transmission. The authors also focused on multi-hop transmission and proved that deploying a single relay would increase significantly the transmission range (from 26.93 to 51.15 m for foggy weather with visibility V = 10 m). Therefore, increasing the number of relays will lead to a better transmission range. The paper provides a very good overview of the capabilities of V-VLC that can be used to check if such a data transmission method is suitable for a certain project.

Digital cameras have become more and more powerful and efficient; therefore, they can be used for more than the usual scope of photography or video. A new communication technique using optical cameras as receivers has been studied in IEEE 802.15 SG7a and is receiving more and more attention in the research field. Optical Camera Communication (OCC) capabilities in outdoor usage feasibility have been analyzed in the paper [98] using a laboratory setup for the experiments. The work focused on observing how the signal quality is affected by fog and proposed a strategy using amplifiers to overcome the issues and to decrease the noise resulting in better performances for an OCC system.

As the paper [94] stated, the future of transportation systems is to focus on information exchange between vehicles and between vehicles and infrastructure. Cooperative driving can be the foundation stone of a better transportation system ensuring higher levels of traffic safety and comfort. Apart from traffic efficiency, the Intelligent Transportation Systems (ITS) will contribute to the goal of achieving autonomous driving. The presented work highlights the fact that VLC is a mature technology for indoor usage, having the potential to be used in outdoor environments in the scope of vehicle-to-vehicle communication. In the survey paper, the authors identified and addressed the open issues and challenges of Vehicular VLC presented in state-of-the-art papers. Such work is very useful for both beginners and experts in the field, providing a detailed view on the V-VLC topic.

## 4. Sensors and Systems for Fog Detection and Visibility Enhancement

Nowadays, vehicles are equipped with plenty of cameras and sensors desired for some specific functionalities that might be used also for fog detection and visibility improvements. For example, Tesla Model S has only for the autopilot functionality 8 surround cameras, 12 ultrasonic sensors, and forward-facing radar with enhanced processing capabilities.

### 4.1. Principles and Methods

ADAS functions are core technologies for actual intelligent vehicles. Applications such as lane marking detection systems or traffic sign recognition [99,100], forward collision warning systems, light detection and ranging, cameras integrated with radars or lidars [10,11,12]—all can be linked to getting useful information that can be further used for visibility detection in fog conditions.

The paper [101] presented a study on how a foggy weather environment can influence the accuracy of machine vision obstacle detection in the context of assisted driving. A foggy day imaging model is described, and the image characteristics are analyzed. The object detection capabilities of machine vision are tested by simulating four types of weather conditions: clear, light fog, medium fog, heavy fog. The study helps in quantifying the effect of fog on machine vision, showing the impact of bad weather conditions on the detection results of assisted driving. Such a study is very helpful to understand how fog can affect an autonomous vehicle or other safety features based on cameras. On the same topic, we can mention [102], where the performances of vision-based safety-related sensors (SRS) are analyzed. The fog was created in a special chamber, and experiments were conducted to determine how the environment is affecting the safety functions. Object recognition tests were conducted using vision-based SRS at low visibility, proposing a method to verify the functional safety of service robots that are using vision-based safety-related sensors. Such work can be extended to test the safety systems presented on a vehicle in foggy conditions.

### 4.2. Onboard Sensors and Systems

Gallen et al. present in [103] a model of reducing the risk of accidents in bad weather conditions due to reducing visibility or friction by computing the advisory speed. This feature can be added to an ADAS system to monitor the speed limit. More than that, different profiles such as emergency breaking, speed profiles based on-road characteristics, etc., are calculated, which together with the vehicle and driver-related parameters (driver’s reaction time, pressure needed to press brake pedal) can be integrated into a single system able to deal with different weather conditions. The model can be further improved by adding other profiles, different driver behavior, and extending to nighttime conditions to assure a complete interaction between driver, car, infrastructure, and environmental conditions.

In [12], Danheim et al. understood the opportunity of using data from different sensors already existing on commercial vehicles but used for other functionalities today, and they described a system for automatic weather conditions recognition composed from a camera and LIDAR. The data from the camera and LIDAR are gathered and interconnected, using a fusion model, to assist the control systems for autonomous driving. For a camera, fog detection has used the approach presented by Pavlic et al. in [75], but the authors think that using just a camera is too risky for autonomous driving, as we concluded at the end of Section 2. The LIDAR, which works on laser technology, can eliminate most of the gaps of the camera approach. It uses the backscattering method, a laser beam is emitted in the atmosphere and comes back to the LIDAR by reflection; in this way, we can calculate the distance to the nearest object. Fog has the effect of an atmospheric veil, and the power of the reflected signal and the backscatter coefficient have been analyzed to determine its presence and then its density. Local weather can be detected near the car, based on some predefined models for every type of reflected signal (ground, fog, rain, snow, etc.), which is information that can be shared with other traffic participants, transforming the vehicle into a reliable weather source. The research continues in [73] where besides fog, air pollution (smog) is detected, which is very useful for big, crowded cities. The detection, realized with the camera, is based on the luminance absorption, reflection, refraction, or scatter of light particles, measurably, due to the molecules and particles from the air. Pollutants cause different levels of absorption, scattering, and reflection in the color spectrum comparing to clean air; that is why analyzing the RGB color channel histogram (Haar wavelet) can provide information related to the degree and type of pollution. LIDAR detection is made, similar to that described above, based on the characteristics of the impulse response—flat and extended for air pollutants, fog, or rain. This last part is very important for the future green eco-driving, which will be the trend for the next few years. Due to the high pollution caused by transportation in the last years, one of the major goals for the automotive industry is to reduce the CO_2_ emissions but also the energy consumption, considering that in 2015, this sector accounted for more than 25% of the global energy, and there is an expected increase of 1.1% every year until 2040 [104]. The first measure to decrease this negative phenomenon was taken by the public authorities; most of the European big cities will not allow the use of diesel vehicles inside the cities starting with 2020.

A radar-based approach is presented in [105], where the scattering and absorption of the millimeter-wave are analyzed; facts are impacted by wavelength, temperature, and particle proprieties. To derive a model for radar reflectivity and then to mathematically link it to visibility, drop size distribution is considered, based on a modified gamma distribution, which is the approach used also in [71]. The drop size distribution parameters allow detecting different fog types and their variation. During the experiments, radar reflectivity is measured by using a 35 GHz cloud radar, visibility with Biral SWS-100 (a visibility sensor with a measurement resolution of 10 m and accuracy of 10%) is able to measure ranges from 10 m to 75 km. The results obtained with the two devices presented above are compared with computed ones based on drop size distribution and information extracted from a Forward Scattering Spectrometer Probe that measured the number of particles from the atmosphere and their size (detected values are between 3 and 46.5 μm). The first measurements prove the relationship between the radar reflectivity and the visibility, the results overlap in different fog moments, some differences appear most due to the device’s limitations—the cloud radar being sensitive to much smaller fog particles compared to FSSP. In this way, a radar-based visibility estimator is developed based on the reflectivity–visibility link, but more data are needed to validate the system.

In degraded visual environments such as fog, the detection of other traffic participants suffers. The paper [106] focuses on a LIDAR target echo signal recognition technology that is based on a multi-distance measurement and deep learning to detect obstacles. There are 2D spectrograms obtained by using the frequency–distance relation divided from 1D echo signals that the LIDAR sensor is providing. The images are analyzed by the proposed algorithms to perform object recognition. Simulation and laboratory experiments were performed to demonstrate the fact that LIDAR detection in bad weather conditions was improved. The experiments performed using a smoke machine to simulate fog show potential in the proposed method (Figure 7), but practical application problems are still to be considered.

Another paper based on laser and LIDAR measurements for visibility distance estimation is presented in [107]. The authors used an experimental laboratory setup (Figure 7) to test and analyze different methods (in similar and repeatable conditions), and the results are compared with the ones obtained from human observers (in the same fog conditions). To determine a mathematical relationship between the laser beam attenuation and the particles’ characteristics, the fog particles were analyzed using a microscope. The last experiment presents a comparison between a LIDAR and a telemeter, proving that the first one can be used in estimating fog conditions. Based on all these results, the authors proposed a collaborative system that gathers data from different sensors and offers more reliable results related to the visibility distance in adverse weather conditions.

Using the setup from Figure 7, the authors realized experiments to estimate visibility distance under fog conditions using laser and LIDAR pieces of equipment. The results of technical measurements performed with those devices were validated by the response of human subjects to the visibility in the same conditions. This type of setup can be used to test any other device in fog conditions, for example, a camera used for image processing.

### 4.3. External Sensors and Systems

A static system proper to be installed on highways to detect the visibility range in fog conditions and to calculate the adequate speed is presented in [108]. The system is composed of a laser and a camera that monitors the length of the beam. In foggy conditions, the laser beam is dissipated, resulting in a shorter trace captured with the camera. Using this information, the visibility range is estimated, and the proper speed is recommended to be able to stop the car in safe conditions. Furthermore, these measurements can be displayed on the highway’s display panels or sent as a notification to the drivers. The measurements and results are the output of laboratory experiments; the system needs to be tested in outdoor conditions to be sure about the validity of the results. A drawback of this system is the high cost: the actual highway infrastructure is not enough to implement this solution; additional devices such as laser and cameras need to be installed.

On the same idea of detecting fog on highways using a static system and warning the drivers, a wireless sensor network is presented in [109] (Table 6). The system consists of wireless sensor terminals, local controller stations, and remote stations. The wireless sensor terminal has the role of a router for relaying signal and includes visibility sensors, building a network of wireless sensors. The external communication between the local station and remote station is realized via 3G modules and satellite modules as a backup solution to guarantee network reliability. Each node gathers information about temperature, humidity, and visibility, and based on these three parameters, the system makes decisions related to the density of the fog. Variable message signs and fog sensors need to be installed at every km to inform the driver about the local weather and the recommended speed, which means very high costs. The results presented in the paper are only preliminary ones, which prove that the process flow is working (data acquisition, transmission, and processing) but there are no results in different weather conditions (no fog, fog, dense fog) presented in the paper to prove the functionality of the system.

A method of estimating fog density based on the free space optical link attenuation is presented in [69] and continued in [70] by using standard meteorological equipment. Brazda et al. present a visibility meter composed of a camera and black and white targets used to measure the contrast. Fog is causing a darkening of the white objects and lighting of the black ones, resulting in a decrease in contrast. The method is based on the visibility definition that says that: “Visibility is a distance x, where the contrast ratio between the apparent contrast measured at a distance x and the intrinsic contrast of the target decreases with 2%”; this statement is considered in the equation:V = ln(0.02)/ln(C(x)/C0) × x,(15)
where V is the visibility, x is the distance between the camera and the target, C(x) is the apparent contrast measured at distance x, and C0 is the intrinsic contrast of the target. The actual contrast is calculated here using the luminance of the black and white targets, being a ratio between their difference and their sum, afterwards calculating the visibility. The results are compared with the ones obtained from official meteorological equipment composed of two visibility sensors (PWD), an optical transmitter, and an optical receiver to measure the attenuation of the optical link on two different channels (CH1 1550 nm and CH2 830 nm). The black and white camera is installed next to the receiver, while the black and white targets (1 × 1 m dimension) are placed next to the transmitter. The distance between them is 60 m. During the experiments, in non-foggy conditions, the contrast between the black and white part of the target was C0 = 0.6. In a foggy environment, the camera measured a visibility distance of 83 m, and the contrast decreases, C(x) = 0.0349, which are results that prove the initial statement, the ratio between apparent and intrinsic contrast being 0.058, which is more than 0.02 than when the fog was assumed to appear. The results show a high correlation between the results obtained with the camera and the ones based on optical attenuation, the system being able to identify even quick changes in visibility (fog density). Comparing with the professional PWD visibility sensors, the presented system can measure visibility along a path rather than only at a fixed point, but it is also cheaper. The drawback of the system is that it cannot be used in night conditions.

Another method for fog detection was proposed by Brazda et al. in [71]; using the same setup presented above, they consider the drop size distribution (DSD) of fog, which was estimated using a modified gamma distribution with three parameters a, b and α—determined based on liquid water content, particle surface area, and visibility (characterized by attenuation). The values for the three DSD parameters, according to ITU-R, are a = 0.027, b = 0.3, and α = 3 for a visibility V = 150 m, which is considered as heavy fog and a = 607.5, b = 3, and α = 6 for a visibility V = 450 m, which is meant as moderate fog. The Particle Volume Monitor PVM-100 measures two moments of DSD: liquid water content and particle surface area. The third parameter of fog, visibility, is determined by measuring the attenuation of the optical link after passing through the fog, using an optical transmitter and receiver. The method is tested in real conditions in a fog-prone area. The results show a large variation of all three parameters, a and α reaching values even out of the expected range. The mean square error is in almost 20% of the cases higher than 0.2. All these reasons led the authors to not recommend an estimation of some typical DSD.

The fog sensor presented in [9,72] measures data about the fog density, temperature, and humidity. The main parameter is the liquid water content (g/m^3^), with it being estimated the fog density from the atmosphere by measuring the attenuation of the light beam in fog conditions. The system is composed of an outdoor unit that sends and receives short infrared pulses, the reflected pulses being influenced by the liquid water content from the environment. The measured data are transferred to an internal unit connected to a PC to be processed. The results are plotted in MATLAB, and fog is determined by applying a threshold on these values. The presented system is not a stand-alone one, the analysis is not done in real time, and the data stored, plotted, evaluated, and conclusions are taken offline. In [110], the system was improved, the setup being many sensors that send information to a central unit and then store it in a database. In addition to liquid water content, humidity, and temperature, a lot of other parameters are monitored and compared with some pre-defined thresholds to identify overflows, in such cases where the system notifies of the problem. All these evaluations are done automatically by the proposed system, in contrast to the methods presented above. The main parameter is the liquid water content; based on it, the visibility is determined as follows:V = 0.024(LWC)^(−0.65) [m].(16)

As future work, the authors plan to combine liquid water content with optical power measurements to get more reliable outputs and to validate the results having two different approaches.

In [111], the authors present a system that measures the size of the fog particle based on the laser diffraction method. The particles were illuminated using a visible laser beam (632 nm and 405 nm) and observed by a digital camera. The growth of the water droplets in fog measured with the presented system was compared with the numerical calculation of the clouds’ water droplets growth, both being exponential functions and growth time being higher for thin fog. The results after the experiments show that the radius of the fog particle varies with the time:r = 3.4 × e^(t/510),(17)
in the case of a water droplet with a mass density of 68 mg/L. The values of the radius, considering the time from 0 to 510 (growth time constant vale of thick fog) seconds presented in the paper, vary from 3.4 to 9.24 μm. The results of the numeric calculation present a radius of the water particle of:r = 12.1 × e^(t/1000),(18)
at a mass density of 7 mg/L, 1000 being the growth time constant value of thin fog. For this interval 0–1000 s, the particle size can vary from 12 to 32.61 μm. In addition to the formulas listed above, there are no other elements and results that sustain the author’s statements, the paper being very poor from an experiments and results point of view. The method still has to be proved with some labor or even real-life experiments in foggy conditions.

Motion detection under non-ideal weather conditions can increase road safety and be used to avoid dangers. The paper [112] proposes an alternative technique to Gaussian-based background modeling to detect and segment moving vehicles. The paper’s goals are achieved using a dynamically adaptive threshold using the full-search sum of absolute difference (FSSAD) algorithm. Therefore, a moving vehicle can be differentiated from a dynamic background.

The paper [113] proposes a method to detect traffic objects in bad weather based on a dual input region-based convolutional neural network. The input consists of infrared and visible images for object detection focusing on thermal and visible proprieties. A traffic surveillance system can be used to send data to a car in a smart city or a smart highway context. The presented method can be used alongside the on-board systems to obtain better performances in detecting traffic objects.

Remaining on static systems, information about weather and the visual condition is needed in the context of traffic safety topics; therefore, a framework to automatically extract data from street-level images is presented by [114]. Deep learning and computer vision are used alongside a unified method that has no pre-defined constraints. Four deep convolutional neural network models, called WeatherNet, are trained to extract weather and visual conditions such as dawn/dusk, day and night, time detection, glare, clear, rainy, snowy, or foggy weather. The framework can have input images or a video stream and shows great performance in extracting valuable information. The novelty of the proposed work consists in its simplicity for practical applications and the fact that there are no pre-defined constraints. WeatherNet can be integrated into smart cities to facilitate autonomous driving or various traffic safety features.

To estimate traffic visibility in nighttime conditions, [115] proposes a Traffic Sensibility Visibility Estimation (TSVE) algorithm that uses laser transmission and image processing. The image processing does not need a reference to the corresponding fog-free images and camera calibration. All the data are collected via static roadside equipment that is analyzed locally or remotely. The current atmospheric transmissivity is calculated based on the laser atmospheric transmission theory. The captured images are analyzed using two image processing algorithms, dark channel prior and image brightness contrast. Performed experiments show that the estimation errors are reduced by such a technique.

Even if it is a difficult task, fog detection from satellite data is approached by [116]. Information from a Meteostat second-generation Spinning-Enhanced Visible and Infrared Imager over the United Arab Emirates was used for the study. An adaptive threshold-based technique using pseudo-emissivity values was implemented to detect nocturnal fog. The methods used reduced the number of false alarms in the fog classification. Such a system can be used just as a support for transport systems.

Fog forecasting and detection are very important when discussing the safety of transportation. Using GEO-KOMPSAT-2A/Advanced Meteorological Imager (GK2A/AMI) alongside auxiliary data, a decision tree fog detection algorithm is proposed by [117]. The goal of the study is to reduce damage caused by fog through real-time fog detection using a high-resolution geostationary satellite. The methods were developed over multiple versions and take into consideration the time of the day (day/dawn/night) and the location (land/sea/coast). Experiments were performed to check if the proposed technique can distinguish fog from low clouds. The performances were always better on land than on the coast and at night than at day (at any location).

## 5. Reaction of Human Subjects

After presenting different methods and systems for visibility enhancement and fog detection plus visibility distance estimation, there remain a few questions: Are these methods and measurements useful for a human being, and how applicable are they in real life? This is from our point of view a big challenge, to build a robust and reliable system for visibility measurement. On the other side, if we refer to the autonomous vehicle, the question will be how they will identify traffic signs or different traffic signaling in bad weather conditions, or which is the visibility limit enacted by these vehicles.

A very interesting experiment related to the influence of luminance and contrast in visual perception is presented in [118]. According to the state-of-the-art studies, humans’ responses to looming are driven by an OFF-mechanism more than an ON-mechanism, which means that humans react better to dark objects seen against a light background than to light objects seen against a dark background. The experiments comprised four stimuli variants, two disk sizes, three presentation times, three extrapolations times, and three conditions of feedback, having in total 216 trials, each repeated twice. During the first experiment (12 participants), only the lightness of the looming object was varied, while for the second experiment (15 participants), both the lightness of the looming object and that of the background were varied. The conclusion after these experiments is that human responses are not affected by changes in lightness and contrast. Indeed, the performance is influence by different regimes of feedback; the reaction time increased when humans receive feedback. This means that the feedback obtained from a system in foggy conditions can be very useful for a driver; having a faster reaction time means that collisions can be avoided. During the experiments, we tested only one level of luminance and one level of contrast; it is necessary to test the system for different levels but also for real-life complex scenarios to prove the reliability of the results.

Considering the driver’s reaction time, which is a parameter also analyzed in the previous method, Tarel et al. in [30] want to prove the usability of a Head-up Display inside the vehicle, which was used to dehaze the images and allow the driver to see behind the fog. The results presented in this work are only the ones obtained in the laboratory, the applicability still needs to be proved in real conditions.

In [70], the method already tackled in Section 4.2 is presented in a system where optical power measurement was combined with contrast variation. The results presented there (see Section 4.2) confirm the theory that the decrease, in contrast, is the main fog effect on visual perception. The authors in [8] present a system that approaches the link between light intensity decrease and visibility decrease in fog conditions. First, they create a concordance between the input power of the light sources (led and laser—most used on modern vehicles) and the output optical power. The input power of the light sources is varied in no fog condition and the output power is measured using an optical receiver; then, the input power is kept constant while the fog is introduced in the setup. All this time, the optical power being monitored. In this way, the fog was split into different categories (low fog, normal fog, and dense fog). For the second step, a camera and an eye chart are introduced in the setup, and different levels of fog are generated. The authors assume that fog can have the same effect as eye diseases, and the authors propose an analogy between different levels of fog and acuity decrease by reading the optotypes from the eye chart using a camera and extracting them with an OCR algorithm. So, after the system gets the data related to optical power from the light assessor, it indicates the fog level, which further is converted in visibility distance. The work continues in [119], where the authors present a more complex laboratory setup with which most of the methods from the state of the art can be tested. Afterwards, it will be necessary to confirm these results in real conditions, on highways, and in complex scenarios inside the cities.

A relationship between radar reflectivity and visibility is proposed in [105]. The reflectivity was measured with a 35 GHz cloud radar and visibility with sensors. The results obtained after these measurements are compared with the ones obtained from a Forward Scattering Spectrometer Probe that gives information related to drop size distribution used also to extract radar reflectivity and visibility. The link between the visibility and the reflectivity (attenuation) is correlated in:V = −(ln(ϵ)/βext),(19)
where V is the visibility, ϵ is the threshold of contrast (normally equal to 0.5), and βext is the extinction coefficient.

## 6. Conclusions

This paper presented methods and systems from the scientific literature related to fog detection and visibility enhancement in foggy conditions that appeared over the past ten years. In the next period, the main focus of the automotive companies will be the development of autonomous vehicles, and visibility requirements in bad weather conditions will be of high importance. The actual methods from the state of the art are based on image processing, optical power measurements, or based on different sensors, some of them already available on actual commercial vehicles but used for different functionalities. The image processing methods are based on cameras, which are devices that have a lot of advantages such as freedom of implementing different algorithms, versatility, or costs, but on the other hand, the results obtained from such a system can be erroneous due to blindness caused by other traffic participants, environment, or weather. Methods based on image processing can be applied for low fog conditions; if fog becomes denser, the system is not able to give any valid output. Some methods presented in the literature work only in day conditions, making them unusable for automotive applications that require systems able to offer reliable results in real time and complex scenarios 24 h/day.

Focusing on the fact that images are degraded in foggy or hazy conditions, the degradation depends on the distance, the density of the atmospheric particles, and the wavelength. The authors in [89] tested multiple single image dehazing algorithms and performed an evaluation based on two strategies: one based on the analysis of state-of-the-art metrics and the other one based on psychophysical experiments. The results of the study suggest that the higher the wavelength within the visible range, the higher the quality of the dehazed images. The methods tested during the experiments were dark channel prior [31], Tarel method [18], Meng method [86], DehazeNet method [87], and Berman method [88]. The presented work emphasizes the fact that there is no method that is superior to every single metric; therefore, the best algorithm would vary according to the selected metric. The results of the subjective analysis revealed the fact that the observers preferred the output of the Berman algorithm. The main conclusion is that it is very important to set the correct expectations that will lead to a selection of some metrics and then, based on that, a dehazing algorithm can be preferred.

Systems based on optical power measurement, by direct transmission or backscattering, improve some of the drawbacks described above for cameras: the result is not influenced by day or night conditions, can measure also very dense fog, and the computational complexity is lower comparing to the previous category, making them more sensitive to very quick changes in the environment, which is important in real-time applications. The results obtained using such systems can be also erroneous, due to environmental conditions (bridges, road curves) or traffic participants; that is why our conclusion after gathering all these methods and systems in a single paper is that at least two different systems shall be interconnected to validate the results of each other.

One big challenge, from our point of view, for the next years in this field is to prove that the results obtained from the systems presented above are valid for a human being. The validity of the results is a relevant topic also for autonomous vehicles that need to identity the road, objects, other vehicles, and traffic signs in bad weather conditions, and the automotive companies shall define the visibility limit for these vehicles.

The evaluation of the state-of-the-art methods is presented in Table 7.

Based on the evaluation criteria listed in the table above (Table 7), we can conclude that a system able to determine and improve visibility in a foggy environment shall include a camera and a device able to make optical measurements in the atmosphere. Both categories have their drawbacks, but putting them together, most of the gaps can be covered; every subsystem can work as a backup and can validate the result offered by the other one. An example can be a system composed of a camera and a LIDAR such as in [12]; both systems are already available on nowadays high-end vehicles, offering reliable results, in real-time, 24 h/day. The results obtained from a vehicle can be shared with other traffic participants from that area, in this way creating a network of systems. The direction of improvement for such a system would be to increase the detection range for LIDARs and to use infrared cameras that can offer reliable results in night conditions and to validate the results obtained from the LIDAR.

This synthesis can be a starting point for developing a reliable system for fog detection and visibility improvement, by presenting the weaknesses of the methods from the state of the art (the referenced articles have more than 30,000 citations in Google Scholar), which can lead to some new ideas of improving them. Additionally, we described ways of interconnecting these systems to get more robust and reliable results.

## 7. Observations and Future Research

The present review focuses on existing solutions that reduce or avoid unfortunate traffic events caused by meteorological phenomena such as fog that drastically reduce visibility. Of high importance are the visibility distance estimation and traffic conditions (the condition of the tires, existing infrastructure, signaling systems, braking systems, etc.). Once the visibility distance is estimated, technical conditions that assure a safe movement of vehicles can be established, taking into consideration the external conditions mentioned above. These will lead to speed adjustment, guarantee a certain response time that is necessary to take decisions by the vehicle’s control systems, and alert the other traffic participants.

As presented in our work, no method exceeds all the metrics when discussing visibility enhancement and fog detection based on image processing. Some methods perform by increasing the contrast of the image, others by increasing the visibility of the edges, while others perform by restoring the image. Therefore, it is clear that using only one camera-based method is not enough to assure the reliability of a safety system installed on a vehicle. The solution could be to use a suite of image processing methods and take from each one only the strong points. Such a system can be considered for future work; in the literature, there are no papers that tried to use multiple algorithms and combine them. Of course, the feasibility of such a system is questionable from different points of view such as computational power, response time, costs, complexity, etc. Another neglected aspect in the literature is combining a camera-based method with other technologies such as Visible Light Communications, LIDAR-based systems, or other infrastructure elements.

In future works, the focus shall be on collaborative systems ([107]) having multiple elements from each field mentioned in our work. First, the external elements placed on the side of the road (smart highways) will estimate the visibility distance and transmit it to the vehicles alongside other data such as traffic volume on the road. Second, internal systems will be designated for object detection, obstacle avoidance, automatic braking, etc. Third, vehicle-to-vehicle and vehicle-to-infrastructure communication shall be integrated (Visible Light Communications). In addition to those systems, satellite-based communication can be used to inform traffic participants regarding the meteorological conditions in large areas (associated with GPS information). The area of using different methods to combat bad weather conditions and increase traffic safety through distance estimation, object detection, vehicle-to-vehicle communication, vehicle-to-infrastructure communication, or image enhancement is still uncharted, and it could represent a field that brings many improvements in the future of safe driving.

The proposed solutions will be in close connection with autonomous vehicles. Due to the dynamics of manifestation, the main challenges for autonomous driving can be separated into two categories. The first category is represented by methods of detecting a vehicle that implies vehicles of different shapes, sizes, or colors, the context in which the vehicle is placed, the surroundings, or other objects that are in the vicinity of the vehicle. On this topic, real-time functionality is of utmost importance, meaning that the vehicle speed can impose limits on the processing speed of used algorithms. A system designed for vehicle detection shall be a robust and reliable one that can cope with situations in which vehicles are moving not only on clear weather but also on heavy rain, fog, snow, etc. The second category is represented by methods of detecting pedestrians and animals. There are different shapes, colors, heights, sizes, and positions that have to be taken into consideration under weather variable conditions. The dynamic situations have to be analyzed and interpreted by the system, therefore imposing a certain response time and robustness to counter unexpected elements.

The question of safety is a crucial factor in the deployment of autonomous vehicles; from a vehicle sensorics point of view, these are the most advanced types of vehicles, as the information necessary for such a vehicle is greatly increased. For now, these solutions have nowhere near peaked their technological potential considering there is still a risk of accidents caused by misinterpretations of the road conditions, thus leading to a reduced level of authorization in most jurisdictions. Unfortunately, the current literature does not feature a study about incidents caused by autonomous vehicles and what might be the root cause, in correlation to the technical equipment of vehicles.

A key element in the validation of the results is a collaborative approach with vehicle manufacturers and the relevant authorities (police, traffic administration, weather centers, vehicle testing, and homologation institutes) as real-world data are essential for any production deployment of such solutions but also in realizing a statistical study of the benefits in deploying the solutions presented.

The authors consider that future research, methods, and technologies shall be oriented toward developing and manufacturing reliable platforms (hardware and software) that comply with existing regulations for example, but not limited to AEC (Automotive Electronics Council), which includes stress and functional safety standards such as AEC-Q100 “Failure Mechanism Based Stress Test Qualification For Integrated Circuits”; AEC-Q101 “Failure Mechanism Based Stress Test Qualification For Discrete Semiconductors”; AEC-Q200 “Stress Test Qualification For Passive Components” and ISO26262 (“Road vehicles—Functional safety”). Succeeding the identification of stress and traffic safety conditions, different design technologies for reliability and fault tolerance must be adopted to obtain a valuable product. A combination of the presented methods together with selection criteria based on the functional scenario would be the most suitable approach for such a task.

## Figures and Tables

**Figure 1 sensors-21-03370-f001:**
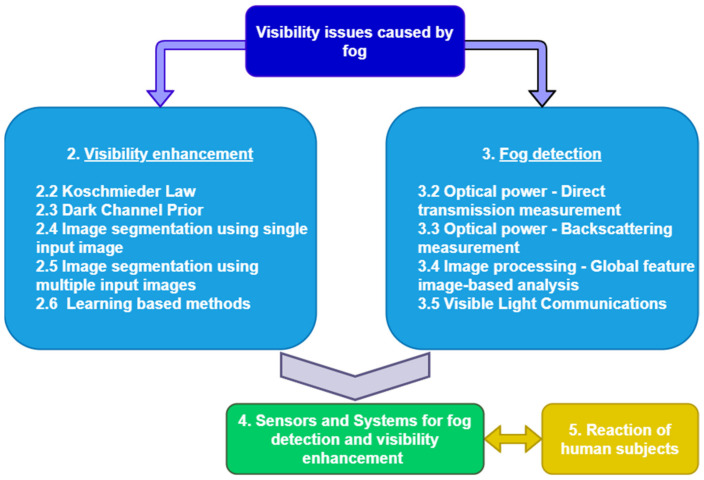
Overall structure.

**Figure 2 sensors-21-03370-f002:**
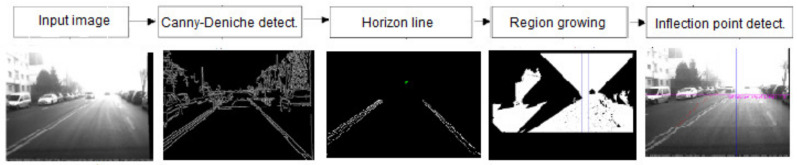
Haze removal method presented in [26].

**Figure 3 sensors-21-03370-f003:**
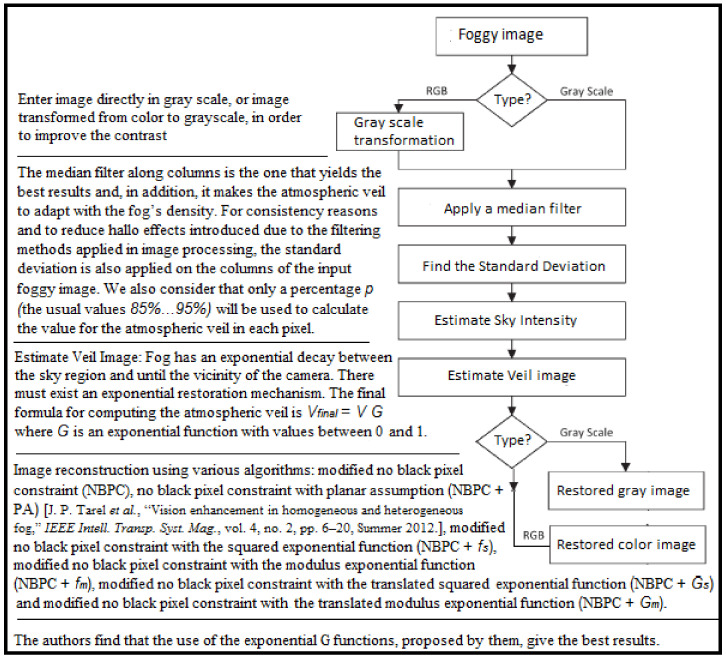
Method proposed in [27].

**Figure 4 sensors-21-03370-f004:**
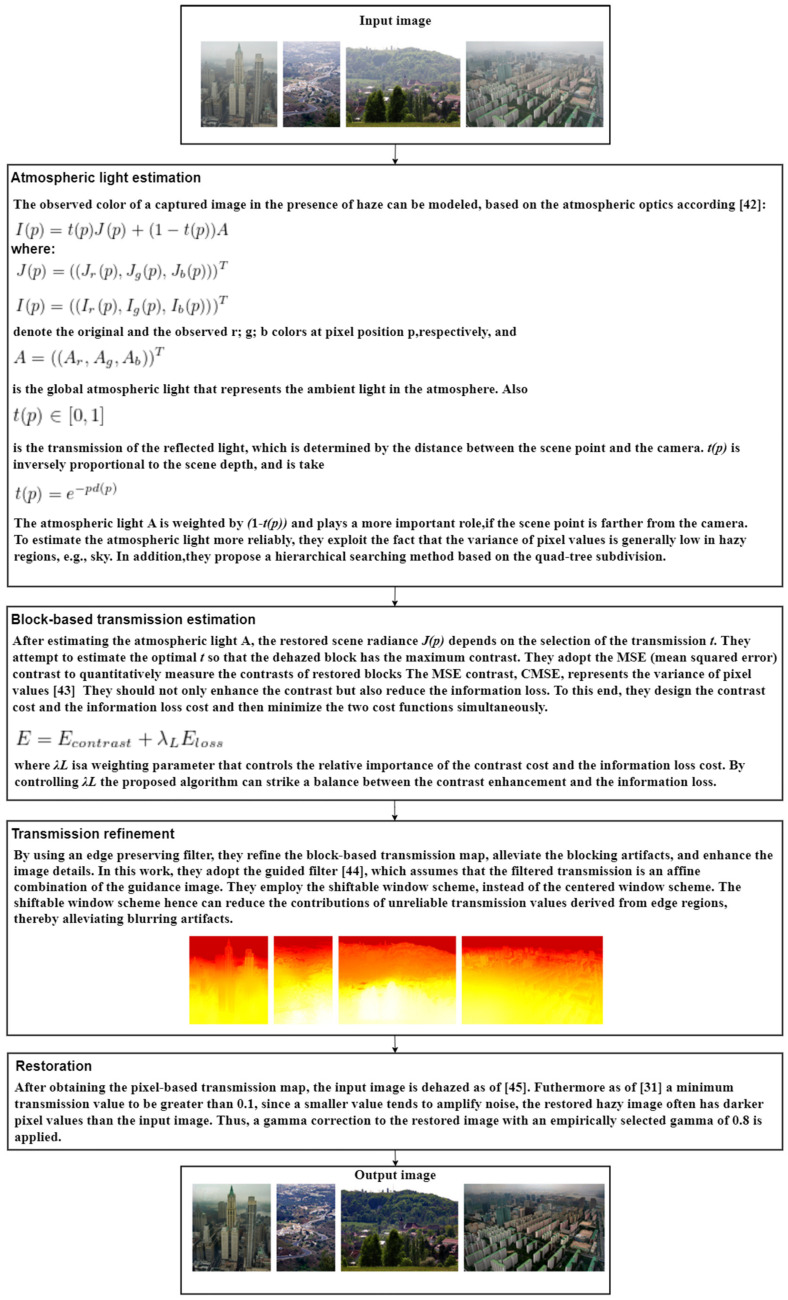
Algorithm description and exemplification of results for the static image dehazing algorithm proposed in [31,41,42,43,44,45].

**Figure 5 sensors-21-03370-f005:**
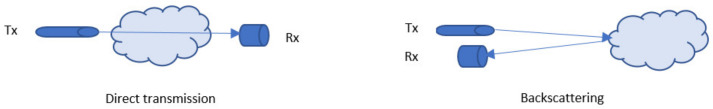
Optical power measurement methods.

**Figure 6 sensors-21-03370-f006:**
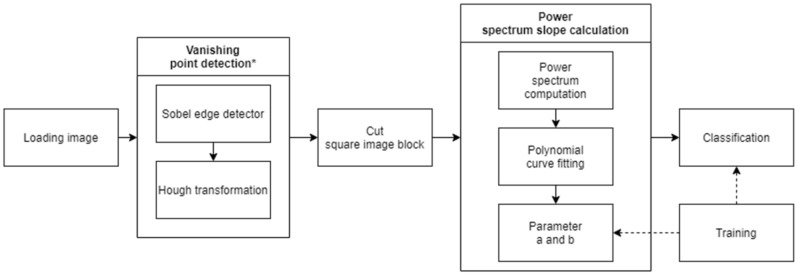
Flowchart of the fog detection algorithm proposed in [77].

**Figure 7 sensors-21-03370-f007:**
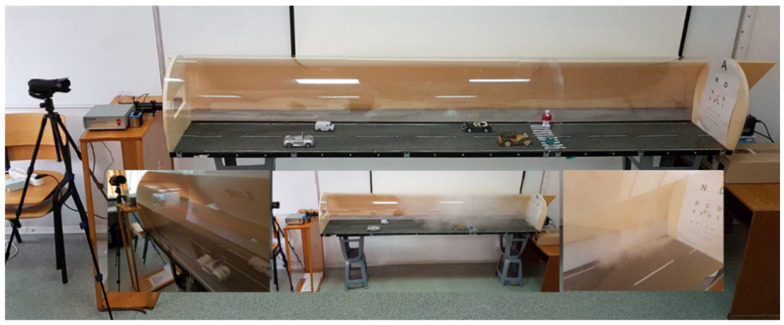
Experimental laboratory setup proposed in [107].

**Table 1 sensors-21-03370-t001:** Method result overview.

Method	Type of Method/Operations	Advantages to Base Solution	Results
Yeh et al. [32]	Addition of two priors pixel-based dark channel prior and pixel-based bright channel prior	Lower computational complexity	Outperforms or is comparable to the reference implementation
Yeh et al. [33]	Addition of two priors pixel-based dark channel prior and the pixel-based bright channel prior	Lower computational complexity	Outperforms or is comparable to the reference implementation
Tan [34]	Markov random fields (MRFs)	Does not require the geometrical information of the input image, nor any user interactions	No comparison to reference made
Fattal [35]	Surface shading model, color estimation	Provides transmission estimates	No comparison to reference made
Huang et al. [36]	Depth estimation module, color analysis module, and visibility restoration	Quality of results increased	Outperforms reference implementation

**Table 2 sensors-21-03370-t002:** Dehazing algorithms comparison from a metrics point of view. Medium haze conditions are considered. The table presents the ranking (from 1 to 5) Data from the table are extracted from [89].

	Algorithm	Dark Channel Prior	Tarel	Meng	Dehaze Net	Berman
Metric						
e Descriptor		2	5	1	4	3
Gray Mean Gradient		1	4	2	5	3
Standard Deviation		1	5	4	3	2
Entropy		1	5	4	2	3
Peak Signal to Noise Ratio		5	3	2	1	4
Structural Similarity Index Measure		5	2	4	1	3

**Table 3 sensors-21-03370-t003:** Dehazing algorithms comparison from human subjects’ point of view. Medium haze conditions are considered. The table presents the ranking (from 1 to 5). Data from the table are extracted from [89].

	Algorithm	Dark Channel Prior	Tarel	Meng	Dehaze Net	Berman
Survey						
Similarity to haze-free image		4	5	1	2	3
Increase in visibility of the objects		2	5	3	4	1

**Table 4 sensors-21-03370-t004:** Overview of VLC patterns and potential applications.

Traffic Elements	Traffic Situations	Possible Events That Shall Be Analyzed from VLC Perspective and the Influence of Weather Factors (Rain, Fog, Smog, Snow)
Infrastructure	Accidents	Unexpected, produce traffic jams by blocking road lanes
	Road junctions	Poorly marked, can contain obstacles that reduce the visibility
	Traffic lights	Faulty functioning, intermittent functioning, not functioning
	Traffic signs	Not functioning, there can be obstacles that reduce visibility
Vehicles in a junction	Head to Head	Faulty signaling
	Head to Tail/ Tail to Head	Safety distance is not kept, headlights or rear lights are not working
	Left side	Can contain obstacles (such as vegetation) that reduce the visibility, traffic rules are not respected because blinkers are not used
	Right side	Can contain obstacles (such as vegetation) that reduce the visibility, traffic rules are not respected because blinkers are not used
Parked vehicles	Parking slots	Moving backwards, sometimes simultaneously with other cars
	Roadside parking	Leaving the parking spot
	Stationary vehicles	In forbidden areas, no warning lights, near junctions or crosswalks
Pedestrians	Jaywalking	Areas with low visibility and no warnings lights
	Exiting vehicle	Areas with high traffic load, getting out of the car without ensuring that there are safe circumstances

**Table 5 sensors-21-03370-t005:** Maximum achievable distance for a reliable transmission for different weather types, BER (bit error rate) = 10^−6^ and V (visibility). The table was constructed using data from [97].

Pulse Amplitude Modulation Size	Maximum Achievable Distance for a Reliable Transmission			
	Clear	Rain	Fog, V = 50 m	Fog, V = 10 m
2-PAM	72.21	69.13	52.85	26.93
8-PAM	53.23	50.98	39.17	19.98
32-PAM	38.73	37.11	28.71	14.66

**Table 6 sensors-21-03370-t006:** Fog detection and warning system setup presented in [109].

Equipment	Components	Communication Link	Roles and Functions
Sensor Terminal	Visibility Sensor/Fog Sensor Wireless Sensor Network Terminal	Wireless sensor network	Collects data from the environment and sends them to the local controller station
Local Controller Station	3G module Satellite module		Processes information from the detector and alerts when pre-defined thresholds are reached
Remote Station		3G and Satellite links	Informs drivers about the visibility conditions in a specific area

**Table 7 sensors-21-03370-t007:** Evaluation of the state-of-the-art methods.

Methods		Evaluation Criteria							
		Computation Complexity	Availability on Vehicles	Data Processing Speed	Day/Night Use	Real-Time Use	Result Distribution	Reliable	Link to Visual Accuracy
Image dehazing	Koschmieder’s law [22,23,24,25,26,27,28,29,30]	Medium/High	Partial (camera)	Medium	Daytime only	Yes	Local for 1 user	No (not for all inputs)	Yes
	Dark channel prior [31,32,33,34,35,36,37,38,39,40,41,42,43,44,45,46,47,48]	High	Partial (camera)	Medium	Daytime only	Yes	Local for 1 user	No (not for all inputs)	Yes
	Dark channel prior integrated in SIDE [49]	High	Partial (camera)	Medium	Both	Yes	Local for 1 user	Yes	Yes
	Image segmentation using single input image [50,51,52,53]	High	Partial (camera)	Low	Daytime only	No	Local for 1 user	No	Yes
	Image segmentation using multiple input images [54,55,56]	High	Partial (camera)	Medium	Daytime only	Yes (notify drivers)	Local for many users (highways)	No (not for all cases)	Yes
	Learning-based methods I [57,58,59,60]	High	Partial (camera)	Medium	Daytime only	No	Local for many users (highways)	Depends on the training data	No
	Learning-based methods II [61]	High	No	Medium	Daytime only	No	Large area	Depends on the training data	Yes
	Learning-based methods III [62,63]	High	Partial (camera)	Medium	Daytime only	No	Local for 1 user	Depends on the training data	Yes
	Learning-based methods IV [64]	High	Partial (camera + extra hardware)	High	Daytime only	Yes	Local for 1 user	Depends on the training data	Yes
	Learning-based methods V [65]	High	Partial (camera)	High	Both	Yes	Local for 1 user	Depends on the training data	Yes
Fog detection and visibility estimation	Direct transmission measurement [8,69,70,71]	Low	No	High	Both	Yes	Local for many users (highways)	Yes	No (still need to prove)
	Backscattering measurement I [9,10,11,12,72,73]	Low	Partial (LIDAR)	High	Both	Yes	Local for 1 or many users	Yes	No (still need to prove)
	Backscattering measurement II [74]	Medium	No	Medium	Both	Yes	Local for 1 or many users	No	Yes
	Global feature image-based analysis [75,76,77,78,79,80,81,82,83,84,85]	Medium	Partial (camera)	Low	Both	No	Local for 1 user	No	Yes
Sensors and Systems	Camera + LIDAR [12]	High	Partial (High-end vehicles)	High	Both	Yes	Local for 1 or many users	Yes	Yes
	Learning based methods + LIDAR [106]	High	Partial (LIDAR)	Medium	Both	Yes	Local for 1 user	Depends on the training data	Yes
	Radar [80]	Medium	Partial (High-end vehicles)	High	Both	Yes	Local for 1 or many users	No (need to be prove in complex scenarios)	Yes
	Highway static system (laser) [108]	Medium	No (static system)	Medium	Both	Yes	Local (can be extend to a larger area)	Yes	No (still need to prove)
	Motion detection static system [112]	Medium	No (static system)	Medium	Day	Yes	Local for 1 or many users	No (not for all cases)	Yes
	Camera based static system [113,114,115]	High	No (static system)	Medium	Both	Yes	Local for 1 or many users	Depends on the training data	Yes
	Satellite-based system I [116]	High	No (satellite-based system)	Medium	Night	Yes	Large area	Yes	Yes
	Satellite-based system II [117]	High	No (satellite-based system)	Medium	Both	Yes	Large area	Yes	Yes
	Wireless sensor network [109]	High	No (static system)	Medium	Both	Yes	Large area	No (not tested in real conditions)	No
	Visibility Meter (camera) [69,70]	Medium	-	Medium	Day time only	No	Local for many users (highways)	No (not tested in real conditions)	No
	Fog sensor (LWC, particle surface, visibility) [71]	Medium	No (PVM-100)	Medium	Both	-	Local for many users (highways)	No (error rate ~20%)	No
	Fog sensor (density, temperature, humidity) [9,72]	Medium	No	Low	Both	No	Local for many users (highways)	No	No
	Fog sensor (particle size—laser and camera) [107,110]	High	Partial (High-end vehicles)	High	Day time only	No	Local for many users (highways)	No	No

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
