# Peer review of "Visibility Enhancement and Fog Detection: Solutions Presented in Recent Scientific Papers with Potential for Application to Mobile Systems"

_sensors, 2021, doi:10.3390/s21103370_

Round 1

Reviewer 1 Report

I. Despite the fact that the literature review presented in the work is extensive, it does not fully exhaust the discussed issue. It is also worth introduce the problems presented in the following articles:

  1.  E. Eso, A. Burton, N. B. Hassan, M. M. Abadi, Z. Ghassemlooy and S. Zvanovec, "Experimental Investigation of the Effects of Fog on Optical Camera-based VLC for a Vehicular Environment," 2019 15th International Conference on Telecommunications (ConTEL), Graz, Austria, 2019, pp. 1-5, doi: 10.1109/ConTEL.2019.8848552.
  2. A. Memedi and F. Dressler, "Vehicular Visible Light Communications: A Survey," in IEEE Communications Surveys & Tutorials, vol. 23, no. 1, pp. 161-181, Firstquarter 2021, doi: 10.1109/COMST.2020.3034224.
  3. Li-da Guo, Ming-jian Cheng, and Li-xin Guo "Visible light propagation characteristics under turbulent atmosphere and its impact on communication performance of traffic system", Proc. SPIE 11170, 14th National Conference on Laser Technology and Optoelectronics (LTO 2019), 1117047 (17 May 2019); https://doi.org/10.1117/12.2534424
  4. X. TIAN, Z. MIAO, X. HAN and F. LU, "Sea Fog Attenuation Analysis of White-LED Light Sources for Maritime VLC," 2019 IEEE International Conference on Computational Electromagnetics (ICCEM), Shanghai, China, 2019, pp. 1-3, doi: 10.1109/COMPEM.2019.8779028.
  5. M. Elamassie, M. Karbalayghareh, F. Miramirkhani, R. C. Kizilirmak and M. Uysal, "Effect of Fog and Rain on the Performance of Vehicular Visible Light Communications," 2018 IEEE 87th Vehicular Technology Conference (VTC Spring), Porto, Portugal, 2018, pp. 1-6, doi: 10.1109/VTCSpring.2018.8417738.
  6. Matus, Vicente; Eso, Elizabeth; Teli, Shivani R.; Perez-Jimenez, Rafael; Zvanovec, Stanislav. 2020. "Experimentally Derived Feasibility of Optical Camera Communications under Turbulence and Fog Conditions" Sensors 20, no. 3: 757. https://doi.org/10.3390/s20030757

II. In addition, the discussion of the results of other papers should be not only done by sentences, but it should be together with the reference to scheme of measurement systems and data processing algorithms discussed in these articles. Without those pictures and graphs, the  article under review is hard to read. Could you may provide more figures and graphs in the body of the paper.

Author Response

Thank you for the review. It was very helpful to improve our work.

Reviewer 2 Report

The authors present a comprehensive review of sensor technology and how it relates to visibility enhancement and fog detection.

Overall, the scale of the review I very good and the authors have provided a further summary of the research to date.

There are three areas where I would like the author to improve the manuscript before it is suitable for publication. First a comprehensive review of the writing is required as there are numerous poorly written sections and grammatical errors throughout the manuscript.

Second, the methodology and justification for the research needs to be improved. In particular, the authors need to provide a more detailed explanation regarding how figure 1 was formulated in order to demonstrate that the review covers all appropriate areas. Secondly the authors need to provide some detail regarding how manuscripts were selected for inclusion in the study

Finally, the authors need to include further referencing when making claims. For example, the first three sentences in the introduction require references to substantiate the claims. There are also further examples throughout the manuscript.

Following substantial editorial review and improvements to the writing. In my opinion the, manuscript should be suitable for publication.

Author Response

(The authors gave the same response as above.)

Reviewer 3 Report

This paper provides a review of visibility enhancement and fog detection. There are some major points that should be addressed to reach the publication level. 

  1. In Figure 1, the number of section 1 seems to be removed.
  2. As a review paper, the authors are expected to provide more classifications in the forms of diagrams, tables, figures, etc. The authors are highly recommended to spend more time on this issue.
  3. What is section "4. Discussion"? is it a brief conclusion or discussion? for the discussion, more interpretations are expected.

Author Response

(The authors gave the same response as above.)

Round 2

Reviewer 1 Report

The text on Figure 4 is nor redable. Please improve it.

Reviewer 2 Report

The authors have addressed my previous comments. The additions improve the manuscript and the writing is clearer following editorial review.

Author Response

Thank you for the review, the observations were valuable to us and lead to the improvement of our work.

Reviewer 3 Report

The authors have made many efforts to improve the quality of the manuscript. So, the paper is acceptable from my point of view. 

Author Response

Thank you for the review, the feedback provided was valuable to us and lead to the improvement of our work.